



# 1 Estimating Radiative Forcing Efficiency of Dust Aerosol
# 2 Based on Direct Satellite Observations: Case Studies over
# 3 the Sahara Desert and Taklimakan Desert

Lin Tian[1,2,3], Lin Chen[3], Peng Zhang[3], Lei Bi[4]
[1] Nanjing University of Information Science & Technology, Nanjing, China.
[2] Chinese Academy of Meteorological Sciences, Beijing, China.
[3] National Satellite Meteorological Center, China Meteorological Administration, Beijing, China.
[4] Department of Atmospheric Sciences, School of Earth Sciences, Zhejiang University, Hangzhou,
China.
*Correspondence to*: Peng Zhang (zhangp@cma.gov.cn) & Lin Chen (chenlin@cma.gov.cn)
**Abstract.** The direct radiative forcing efficiency of the dust aerosol ($DRFE_{dust}$) is an important indicator
to measure the climate effect of the dust. The $DRFE_{dust}$ is determined by the microphysical properties
of the dust, which vary with the dust source regions. However, there are only sparse in-situ
measurements of them, such as the distribution of the dust aerosol particle size and the complex
refractive index in the main dust source regions. Furthermore, recent studies have shown that the
non-spherical effect of the dust particle is not negligible. The $DRFE_{dust}$ is often evaluated by estimating
given microphysical properties of the dust aerosols in the radiative transfer model (RTM). However,
considerable uncertainties exist due to the complex and variable dust properties, including the complex
refractive index and the shape of the dust. The $DRFE_{dust}$ over the Taklimakan Desert and the Sahara
Desert is derived from the satellite observations in this paper. The advantage of the proposed
satellite-based method is that there is no need to consider the microphysical properties of the dust
aerosols in estimating the $DRFE_{dust}$. For comparison, the observed $DRFE_{dust}$ is compared with that
simulated by the RTM. The differences in the dust microphysical properties in these two regions and
their impacts on $DRFE_{dust}$ are analyzed.
The $DRFE_{dust}$ derived from the satellite observation is $-39.6 \pm 10.0 \ \mathrm{Wm^{-2}\tau^{-1}}$ in March 2019 over
Tamanrasset and $-48.6 \pm 13.7 \ \mathrm{Wm^{-2}\tau^{-1}}$ in April 2019 over Kashi. According to the analyses of their
microphysical properties and optical properties, the dust aerosols from the Taklimakan desert (Kashi)
scatter strongly. The RTM simulated results ($-41.5$ to $-47.4 \ \mathrm{Wm^{-2}\tau^{-1}}$ in the Taklimakan Desert and
$-32.2$ to $-44.3 \ \mathrm{Wm^{-2}\tau^{-1}}$ in the Sahara Desert) are in good agreement with the results estimated by
satellite observations. According to previous studies, the results in this paper are proved to be



reasonable and reliable. The results also show that the microphysical properties of the dust can
significantly influence the $DRFE_{dust}$. The satellite-derived results can represent the influence of the dust
microphysical properties on the $DRFE_{dust}$, which can also validate the direct radiative effect of the dust
aerosol and the $DRFE_{dust}$ derived from numerical model more directly.
**1 Introduction**
Dust aerosols are considered to be one of the major components of the tropospheric aerosols
(Huneeus et al., 2012;Textor et al., 2007). The dust aerosols affect the radiation balance of the
earth-atmosphere system by scattering and absorbing solar radiation directly (Miller et al.,
2014;Satheesh, 2002). Estimating the direct radiation effect of the dust aerosol ($DRE_{dust}$) is crucial for
estimating climate forcing. The scattering of the dust influences the radiation in the shortwave (SW)
spectrum at the top of atmosphere (TOA), which causes stronger SW $DRE_{dust}$ over dust source regions
(Slingo et al., 2006). Therefore, the evaluation of SW $DRE_{dust}$ is important for climate modeling.
The variabilities of the mineral dust composition from soils in different source regions cause the
differences in dust microphysical properties (e.g., refractive index, size, and particle shapes). Anderson
et al. (2005) defined the Direct Radiative Forcing Efficiency of the dust aerosol ($DRFE_{dust}$) to quantify
the dust radiative effect (Anderson et al., 2005). The $DRFE_{dust}$ represents the $DRE_{dust}$ of a certain
aerosol optical depth (AOD) at per unit area, which means the efficiency of the dust aerosol that affects
the net radiative flux of solar radiation. The $DRFE_{dust}$ is largely determined by the optical properties of
the dust aerosols (Shi et al., 2005), which are strictly controlled by the microphysical properties of the
particles (Di Biagio et al., 2014b;Di Biagio et al., 2017;Di Biagio et al., 2014a;Zhang et al., 2006).
Therefore, the $DRFE_{dust}$ is different concerning the dust aerosols from different source regions (Tanré
et al., 2001;Che et al., 2012). Without considering the influence of the aerosol loading on the $DRE_{dust}$,
the $DRFE_{dust}$ has unique advantages in evaluating the differences of dust microphysical properties and
their impacts on the $DRE_{dust}$ from different dust source regions (García et al., 2008).
The $DRFE_{dust}$ is often estimated by the General Circulation Model (GCM) and the Radiative
Transfer Model (RTM). Many studies have simulated the SW $DRFE_{dust}$ in different regions
(Valenzuela et al., 2012;Che et al., 2009;Bi et al., 2014). However, there are sparse in-situ
measurements of the dust microphysical properties in the main source regions. The large spatial



variability of aerosols and the lack of an adequate database on their properties makes $DRE_{dust}$ and
$DRFE_{dust}$ much difficult to estimated (Satheesh and Srinivasan, 2006). To date, climate models
generally use temporal and spatial constant values to represent the dust microphysical properties (Di
Biagio et al., 2017;Di Biagio et al., 2014a;Bi et al., 2020). This may cause uncertainties in calculating
the dust radiative effect. Moreover, the shape of the dust particle in the model needs to be assumed.
Therefore, there are large uncertainties in estimating the $DRFE_{dust}$ with few measurements of the dust
microphysical properties from different source regions (Bi et al., 2020;Colarco et al., 2014;Zhao et al.,

66  2013).

Satellite observations can be used in estimating the $DRFE_{dust}$ because satellites can directly
observe the radiation budget of the earth in the TOA (Wielicki et al., 1998;Satheesh and Ramanathan,
2000), and the remote-sensing technique for the AOD has been developed (Remer et al., 2005;Hsu et
al., 2004). In the previous study, we developed a satellite-based method to estimate the $DRFE_{dust}$ over
land without any assumptions of the microphysical properties of dust aerosols (Tian et al., 2019). In
previous researches, performances of the models in simulating the dust radiative effect have been
indirectly validated by comparing the observations of the AOD, the single scattering albedo (SSA), the
distribution of the particle size, and the extinction profile of the aerosols with the simulated ones (Zhao et
al. 2010; Chen et al. 2014). Therefore, the satellite-based method provides a direct way to validate the
$DRE_{dust}$ and the $DRFE_{dust}$.
The Taklimakan Desert and the Sahara Desert are the main dust source regions, which influence
many areas (Li et al., 2020;Mikami et al., 2006;Mbourou et al., 1997;Huang et al., 2014). Thus, the
assessment of the SW $DRFE_{dust}$ and microphysical properties of the dust over these regions is
important for evaluating regional and global climate changes.
In this paper, the $DRFE_{dust}$ in dust storms over the Taklimakan Desert and the Sahara Desert is
evaluated based on satellite observations and the RTM, separately. With the comparison of the dust
microphysical properties and the $DRFE_{dust}$ in these two regions, the differences of the dust
microphysical properties are analyzed. Meanwhile, the influences of the dust microphysical properties
on the $DRFE_{dust}$ are investigated in this paper. The need for accurate information on the dust
microphysical properties and dust sources for simulating the $DRFE_{dust}$ is emphasized, and the
advantage of the satellite-based method in estimating the $DRFE_{dust}$ is revealed.





**2 Methodology and data**

In the previous study (Tian et al., 2019), the equi-albedo method has been proposed to estimate the $DRE_{dust}$ and the $DRFE_{dust}$ over land based on satellite measurements directly. This method bases on the assumption that the SW radiative fluxes at the TOA of the clear sky ($F_{clr}$) are equal over the regions with similar land surface albedo (LSA) and solar zenith angle (SZA). Following this method, we estimated the $DRFE_{dust}$ based on the AOD and the SW radiative flux product from the same satellite platform.

Moreover, the $DRFE_{dust}$ in the RTM with dust aerosol microphysical properties is also evaluated. Based on the comparison between the $DRFE_{dust}$ results from the two methods, the differences in the dust microphysical properties over the Taklimakan Desert and the Sahara Desert are analyzed, and the differences in the $DRFE_{dust}$ are also discussed. The processing steps are shown in Fig. 1.

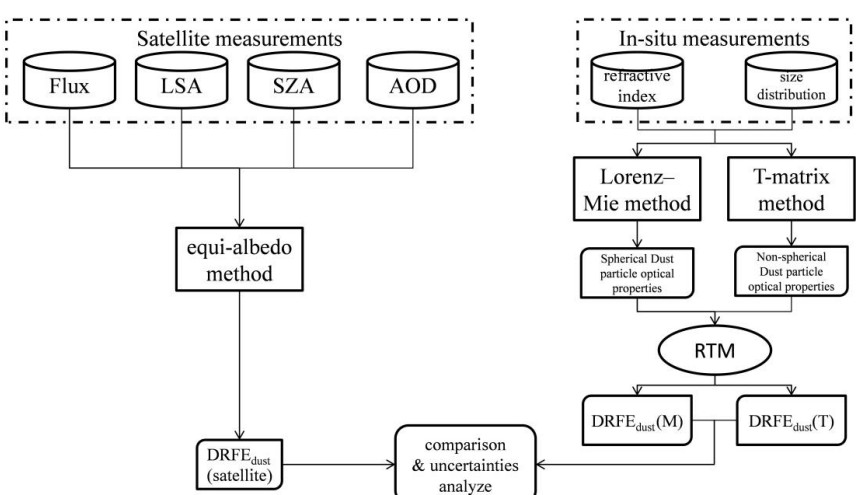

Flux: Radiative Flux observed by CERES;
LSA: Land Surface Albedo;
SZA: Solar zenith Angle;
AOD: Aerosol Optical Depth;
$DRFE_{dust}$(satellite): $DRFE_{dust}$ estimated from satellite measurements
$DRFE_{dust}$(M): $DRFE_{dust}$ simulated from Lorenz–Mie method and RTM;
$DRFE_{dust}$(T): $DRFE_{dust}$ simulated from T-matrix method and RTM;

**Figure 1: Processing flow chart of this paper.**



## 2.1 Methodology

### 2.1.1 The equi-albedo method

Previous studies have shown that $F_{clr}$ is significantly influenced by the LSA and the SZA at the TOA (Di Biagio et al., 2012;Tegen et al., 2010). It is hard to assess the SW $DRE_{dust}$ and the $DRFE_{dust}$ over land derived from satellite observations due to the large dynamic range of the LSA (Satheesh, 2002). In the previous study (Tian et al., 2019), we proposed an equi-albedo method to minimize the influence of the inhomogeneous LSA and SZA and directly derived the $DRE_{dust}$ and the $DRFE_{dust}$ over land from satellite observations based on the assumption that the $F_{clr}$ is equal over the regions with similar LSA and SZA.

$DRE_{dust}$ was defined as the radiative fluxes difference between clear ($F_{clr}$) and dust loading ($F_{dust}$) conditions (Garrett and Zhao, 2006;Christopher et al., 2000;Ramanathan et al., 1989).

$$DRE_{dust} = F_{clr} - F_{dust} \qquad (1).$$

$F_{dust}$ is the shortwave radiative flux at TOA in the cloud-free and dust aerosol loading condition which is obtained directly from CERES data, and $F_{clr}$ is the shortwave flux over the same region without aerosol. $F_{clr}$ cannot be observed directly, and the estimating of $F_{clr}$ must be on the basis of some realistic assumptions.

The equi-albedo method bases on the assumption that the SW radiative fluxes at the TOA of the clear sky ($F_{clr}$) are equal over the regions with similar land surface albedo (LSA) and solar zenith angle (SZA). Based on the assumption, the $F_{clr}$ were estimated, then $DRE_{dust}$ can be derived following Eq. (1). According to the definition of $DRFE_{dust}$, it represents the net flux of solar radiation perturbed by per unit dust AOD. Therefore, $DRFE_{dust}$ can be expressed as:

$$DRFE_{dust} = DRE_{dust}/\tau_{dust} \qquad (2)$$

where $\tau_{dust}$ is the AOD of dust aerosols, and $\tau_{dust}$ comes from the MODIS aerosol product. Thus, $DRFE_{dust}$ was estimated based on the AOD and the SW radiative flux product from the same satellite platform.

In the previous study (Tian et al., 2019), we have estimated the $DRE_{dust}$ and the $DRFE_{dust}$ of two dust storms in the Taklimakan Desert. The results were compared with the $DRE_{dust}$ and the $DRFE_{dust}$ simulated by the RTM. The results indicated that the method is effective in estimating the SW $DRFE_{dust}$ over land. The microphysical properties of dust aerosols significantly influence on the $DRE_{dust}$ and the $DRFE_{dust}$ (Che et al., 2012;Li et al., 2018). The different microphysical properties of dust aerosols in



various dust source regions cause uncertainties in estimating the SW $DRE_{dust}$ and $DRFE_{dust}$. Thus, the
equi-albedo method is used to estimate the SW $DRFE_{dust}$ directly using satellite observations in this
study. Based on the comparison of the $DRFE_{dust}$ in the Taklimakan Desert and the Sahara Desert, the
differences of dust microphysical properties in these two regions are analyzed and the influences of the
dust microphysical properties on estimating the $DRFE_{dust}$ are tested.

**2.1.2 Calculating method of dust optical properties**

Dust aerosols are often assumed as spherical particles in the GCM and the RTM (Wang et al.,
2013;Gao and Anderson, 2001). The Lorenz-Mie theory is used to calculate the optical properties of the
dust particles (Gouesbet and Gréhan, 2011). However, observations and researches have shown that
most dust aerosols are non-spherical in nature (Nakajima et al., 1989;Okada et al., 2001). Previous
researches also suggested that assuming particles as spherical or non-spherical has significant impacts
on calculating the dust optical properties (Kalashnikova and Sokolik, 2004;Borghese et al., 2007).
Therefore, the optical properties of dust aerosols are calculated using both the spherical and the
ellipsoidal methods for comparison to analyze the uncertainties caused by the assumption of dust
shapes in estimating the $DRFE_{dust}$ in this study.
To make it more accurate, the light scattering properties of spherical particles are generally
calculated based on the Mie and Lorenz theory (Mishchenko and Travis, 2008). Among several
methods for computing optical properties of non-spherical particles, the T-matrix method has been
extensively developed to many versions for various applications (Chylek et al., 1977;Mishchenko et al.,
1996). These versions of the available T-matrix code are accessed from the National Aeronautics and
Space Administration (NASA) Goddard Institute for Space Studies (GISS) group (Mishchenko and
Travis, 1998). Mie scattering method can be regarded as a special case of the T-matrix method. In this
study, the NASA-GISS code is used to calculate the optical properties of the spherical particles and the
ellipsoidal particles. The particle aspect ratio is set to 0.8.

**2.1.3 RTM**

Santa Barbara Disort Atmospheric Radiative Transfer (SBDART) is an RTM that calculates the
plane-parallel radiative transfer of the earth-atmosphere system (Ricchiazzi et al., 1998). The broadband
radiative flux at the TOA and the surface in clear-sky and dusty conditions can be obtained. It is



conducive to analyzing the radiative transfer theory in satellite remote sensing and atmospheric energy
budget studies. Furthermore, the model can flexibly set up aerosol properties, which is well suited to
calculate the radiative effect of different types of aerosols. The SBDART model has been widely used in
estimating the $DRFE_{dust}$ due to its design (Chen et al., 2011;Li et al., 2020;Iftikhar et al., 2018).
In this paper, the dust aerosol optical properties (the SSA and the ASYmmetry parameter,
abbreviated as ASY) are calculated using spherical and non-spherical methods. The Aerosol Robotic
Network (AERONET) inversion products, the LSA from Moderate Resolution Imaging
Spectroradiometer (MODIS) surface albedo product, and the default atmospheric profile of SBDART
(MID-LATITUDE WINTER) are used as the input parameters for the SBDART model in simulating the
$DRFE_{dust}$. Therefore, the $DRE_{dust}$ changing with the AOD due to both dust aerosol microphysical
properties (including the complex refractive index and the distribution of the size) and optical properties
(including the SSA and the ASY) are simulated by the SBDART model. The impacts of the
microphysical properties and the optical properties of the dust aerosol on the $DRE_{dust}$ are analyzed in
this study.
**2.2 Data**
This paper aims to analyze the differences in dust microphysical properties and the $DRFE_{dust}$ over
the Taklimakan Desert and the Sahara Desert to confirm the influences of dust aerosol microphysical
properties on simulating the $DRFE_{dust}$. Also, the advantages of the satellite-based method in estimating
the $DRFE_{dust}$ are analyzed. Therefore, the $DRFE_{dust}$ over the Taklimakan Desert and the Sahara Desert is
estimated by using both satellite observations and dust microphysical properties.

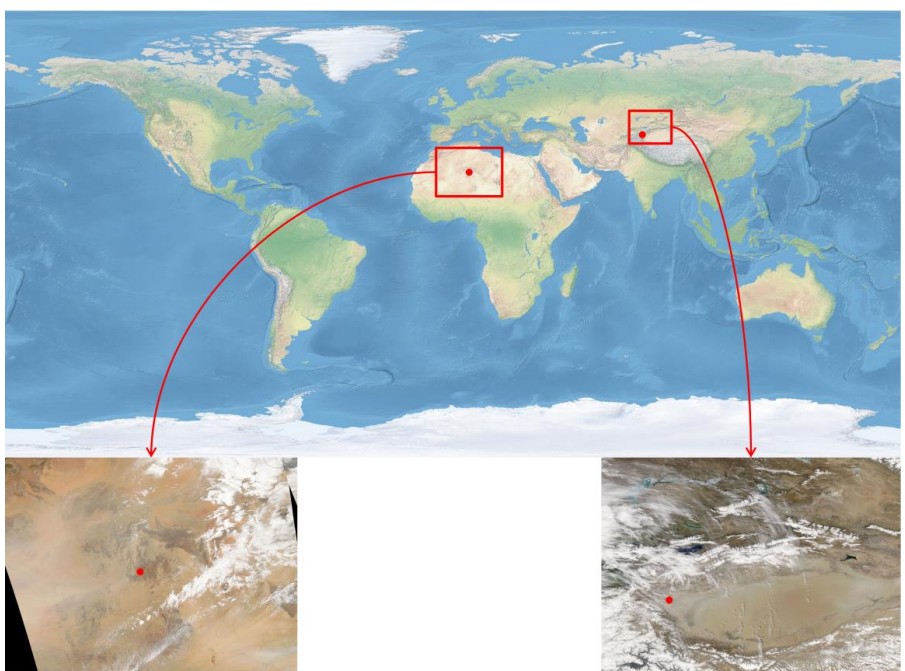


**Figure 2: The research regions and dust storms viewed by MODIS Aqua on 11 March and 9 April 2019.**

Fig. 2 shows the research regions (the red square areas) and the locations of in-situ sites

(Tamanrasset site and Kashi site, the red dots in the map and satellite images). Tamanrasset (22.79˚N,
5.53˚E, 1377 m above the mean sea level) locates in southern Algeria, which is free from the influence
of industrial activities. Thus, the aerosols measured in Tamanrasset can represent the pure dust aerosols
from the Sahara Desert (Guirado-Fuentes et al., 2014). Kashi (39.5˚N, 75.9˚E, 1320 m above the mean
sea level) locates in the vicinity of the Taklimakan Desert. Kashi represents a place affected by dust
aerosols transported from the Taklimakan Desert (Li et al., 2020). Thus, dust aerosols observed in
Tamanrasset and Kashi sites are typical samples of the dust aerosols from these two deserts. Moreover,
Tamanrasset and Kashi sites are similar in land surface type, altitude, and climate. As the LSA and the
SZA have a great impact on the SW radiative effect, the regions with similar LSA and SZA are chosen to
avoid the influence of different LSA and SZA on evaluating the differences of dust microphysical
properties and dust radiative effect from different dust source regions.

A dust storm occurred on 11 March 2019 in Tamanrasset. In Kashi, a dust storm occurred on 9

April 2019. These dust storms are shown visually by Aqua MODIS (Fig. 2). Fig. 3 shows the LSA and
the SZA observed by the AQUA satellite on 11 March 2019 in the Sahara Desert and on 25 April 2019



in Taklimakan Desert. In Fig. 3, the LSA and the SZA are similar in Tamanrasset and Kashi when the
satellite passes through. The data around Tamanrasset and Kashi in March and April are suitable for
analyzing the differences of dust microphysical properties and their influences on the DRFE$_{dust}$.

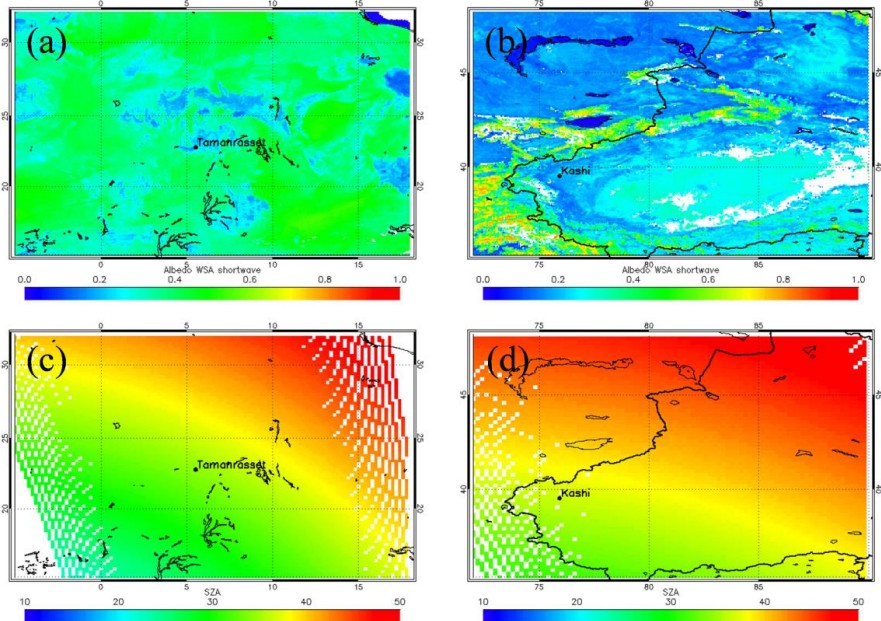

**Figure 3: (a) MODIS SW LSA and (c) SZA on 11 March 2019 over Tamanrasset; (b) MODIS SW LSA and**
**(d) SZA on 24 April 2010 over Kashi.**

The satellite-observed and dust microphysical properties data of the dust storms in March and

April 2019 in Tamanrasset and Kashi are collected to analyze the dust microphysical properties and
estimate the DRFE$_{dust}$ in the Taklimakan Desert and the Sahara Desert. Fig. 4 shows the satellite images
of these dust storms, which can be seen from the satellite images in cloud-free conditions over
Tamanrasset (left column of Fig. 4, Fig. 4(a), Fig. 4(c), Fig. 4(e)) and Kashi (right column of Fig. 4,
Fig. 4(b), Fig. 4(d), Fig. 4(f)). Both the satellite data and synergy dust microphysical properties data are
collected around Tamanrasset and Kashi sites for analyzing the differences in dust microphysical
properties and estimating the DRFE$_{dust}$.

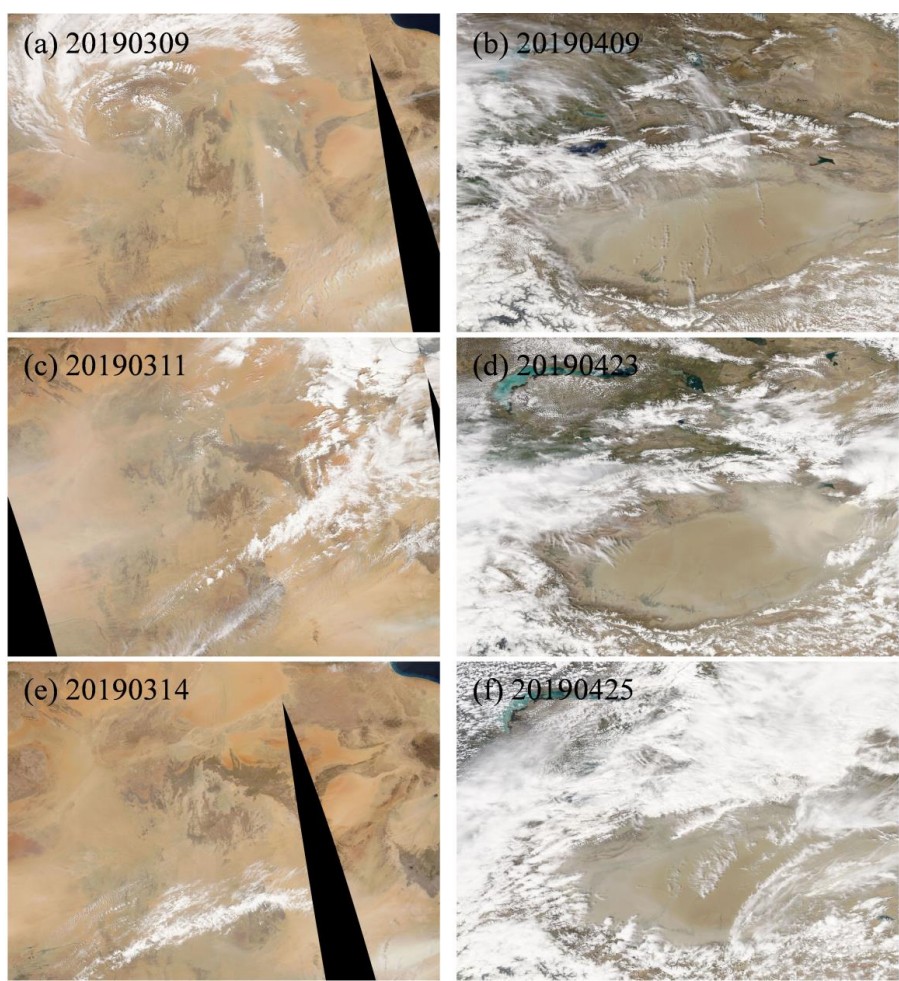

**Figure 4: Dust storms viewed by AQUA/MODIS over target areas (Tamanrasset and Kashi).**
**2.2.1 Satellite data**
MODIS and CERES (Clouds and the Earth's Radiant Energy System) are the key instruments of
the AQUA and the TERRA satellite and are important in NASA's Earth Observing System (EOS). The
AOD products from MODIS and the radiative flux products at the TOA from CERES can be
synergistically used to estimate the $DRFE_{dust}$ directly.
Several algorithms have been developed for MODIS AOD remote-sensing products after MODIS
instruments were launched (Remer et al., 2005). Of these algorithms, the Deep Blue algorithm (Hsu et
al., 2004) solved the problems in aerosol retrieval by satellite remote-sensing for high reflectance land
surface types (such as arid, semi-arid, and desert areas), and retried the AOD over high reflectance land





surface types. In this paper, the deep blue AOD (0.55μm) data are used to discriminate the dust storm
regions. The LSA is also needed both in the satellite-based equi-albedo method and the RTM. The
MODIS Collection6 albedo product dataset (MCD43C3) (Schaaf et al., 2011;Schaaf et al., 2002;Schaaf
et al., 2008) provides high-quality land surface reflectance and albedo data over various types of land
surfaces by using anisotropy retrievals algorithm (Jin et al., 2003;Liang et al., 2002;Liu et al.,
2009;Román et al., 2010). The MCD43C3 product dataset is available from the Land Processes
Distributed Active Archive Center (LP DAAC) of NASA. The white-sky albedo from the MCD43C3
product is used to get the SW broadband LSA.
CERES single scanner footprint (SSF) level 2 dataset can provide the radiative flux at the TOA in
three broadband channels. Here the instantaneous SW channel (0.3–5.0 μm) radiative flux at the TOA
from CERES SSF level 2 dataset is used. MODIS and CERES are onboard in the same satellite
platform (AQUA). The radiative flux derived from CERES is co-located with the MODIS scene. The
$DRE_{dust}$ and the $DRFE_{dust}$ at the TOA are estimated by synergistically using MODIS and CERES
products.
**2.2.2 Dust microphysical properties data**
The Aerosol Robotic Network (AERONET) is the largest ground-based network for measuring
aerosols with more than 400 sites installed.
The AERONET provides microphysical properties and optical properties of the aerosols at four
wavelengths (440, 675, 870, and 1020 nm). The AOD product is directly measured by the sun
photometer. The inversion algorithm retrieves the physical properties of aerosols such as volume size
distributions and the complex refractive index, and optical properties such as the SSA and the ASY
(Dubovik and King, 2000;Dubovik et al., 2006).
**3 DRFE$_{dust}$ estimated based on satellite observations**

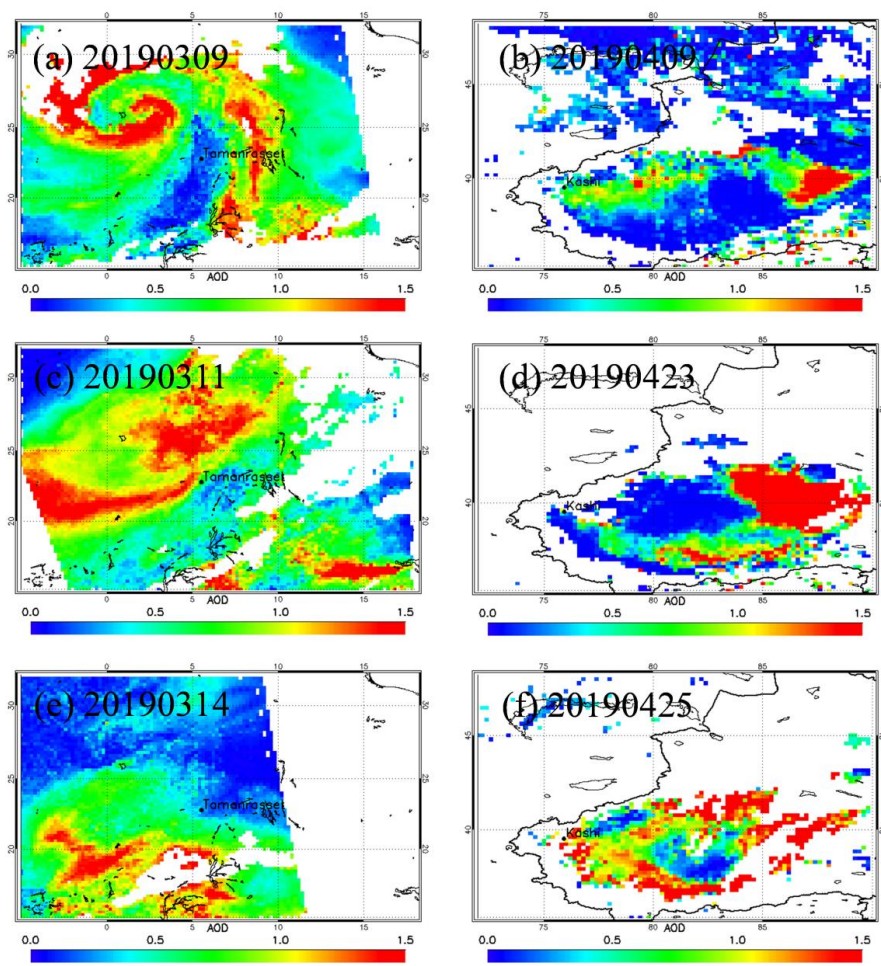


**Figure 5: AOD at 0.55 μm ($\tau_{550}$) of the dust storm in March 2019 over Tamanrasset and that in April 2019**
**over Kashi.**

MODIS L2 deep blue AOD product of the dust storm in March 2019 over Tamanrasset and that in

April 2019 over Kashi are shown in Fig. 5. The missing data are shown in white; the high dust loading
regions are shown in red; the low dust loading regions are shown in blue. Fig. 5 shows that there are
heavy dust storms over Tamanrasset and Kashi with AOD great than 1.0 detected by MODIS.

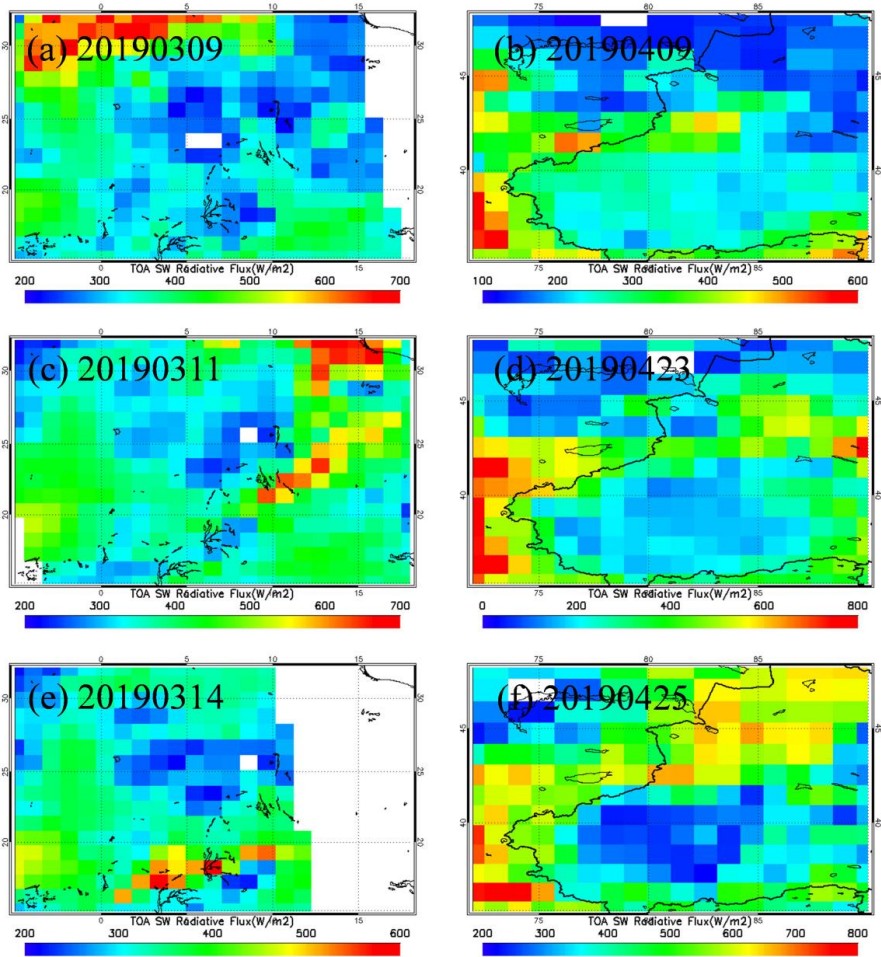

**Figure 6: TOA SW radiative flux in March 2019 over Tamanrasset and that in April 2019 over Kashi.**
Fig. 6 shows the TOA SW radiative flux measured by CERES in March 2019 over Tamanrasset
and that in April 2019 over Kashi during the dust storms. The TOA SW radiative flux distribution
shows the highest value over cloud conditions. The values in dust storm regions are higher than those
in clear-sky regions. It is due to the fact that the SW albedo of the aerosols in the cloud and the dust is
higher than those on the land surface. Thus, dust aerosols have a negative radiative effect in the SW
spectrum. Following the equi-albedo method (Tian et al., 2019), the DRE$_{dust}$ is estimated based on the
measurements from MODIS and CERES both aboard on the AQUA satellite.
As Fig. 4, Fig. 5 and Fig. 6 shown, the spatial resolution of TOA flux from CERES/SSF product
is 1°×1 °grid, and LSA, SZA, AOD data from satellite have the different spatial resolution. In order to





match up LSA, SZA and AOD data with CERES TOA SW fluxes, we have resampled LSA, SZA and
AOD data into CERES SSF product horizontal spatial resolution. Then the $F_{clr}$ and $DRE_{dust}$ over
Tamanrasset and Kashi can be estimated following equi-albedo method.

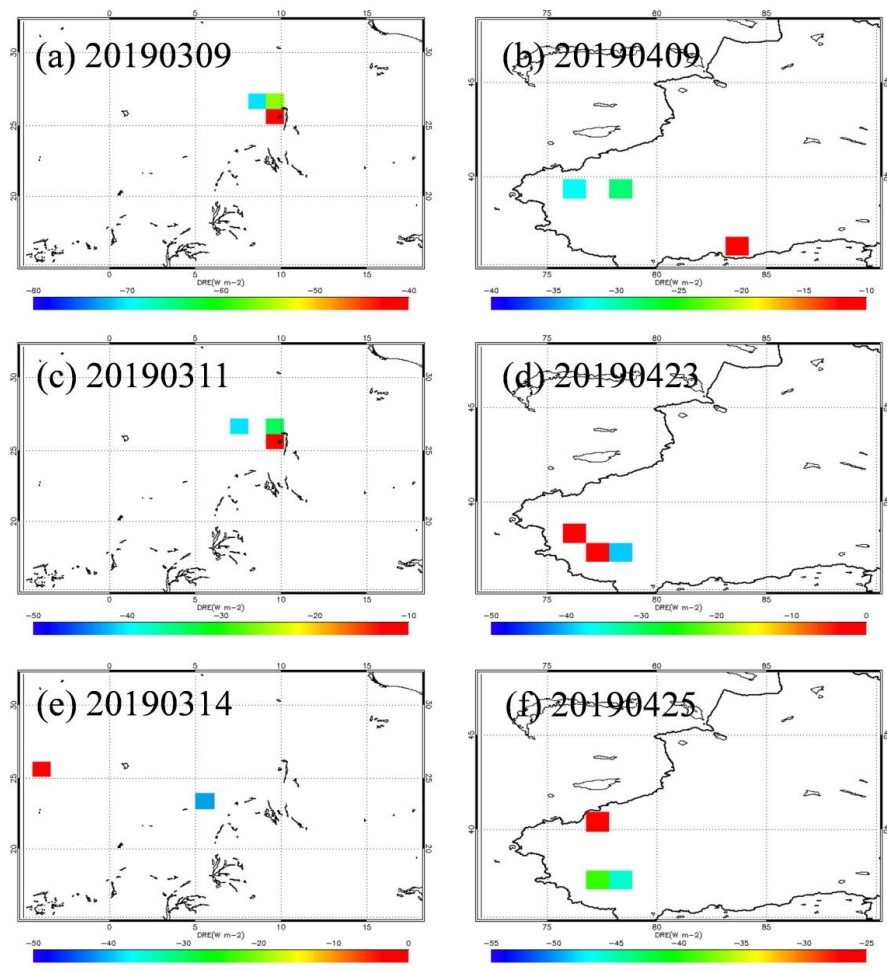


**Figure 7: $DRE_{dust}$ on March 2019 over Tamanrasset and on April 2019 over Kashi.**

Fig. 7 shows the distribution maps of the $DRE_{dust}$. The high dust aerosol loading regions show

significant negative radiative forcing. It indicates that the dust aerosol loading is negatively correlated
with the $DRE_{dust}$ in these dust storm events. The distribution maps of the LSA and the SZA (Fig. 3)
show that the mean SW LSA measured by MODIS is around 0.18 and the mean SZA is around 35
degrees in Tamanrasset and Kashi. The distribution maps also show that the LSA and SZA vary greatly
in the same satellite scan image. To avoid the influence of the LSA and SZA in estimating the $DRFE_{dust}$,





pixels with LSA of 0.16–0.20 and SZA of 32–38 degrees are chosen to derive the $DRFE_{dust}$. Therefore,
only few pixels having similar values of the LSA and the SZA over Tamanrasset and Kashi are picked
for estimating the $DRE_{dust}$ and the $DRFE_{dust}$. The influences of the dust microphysical properties on the
$DRFE_{dust}$ are investigated. These pixels of the $DRE_{dust}$ and its co-located AOD values are illustrated in
Table 1.
**Table 1: $DRE_{dust}$ and AOD in March 2019 over Tamanrasset and that in April 2019 over Kashi during the**
**dust storms.**

| Regions & Dates | Properties | AOD | $DRE_{dust}$ |
|---|---|---|---|
| Sahara Desert | 20190309 | 0.92 | -41.2 |
| | | 1.51 | -63.7 |
| | | 1.11 | -57.8 |
| | | 0.31 | -15.6 |
| | 20190311 | 0.48 | -11.5 |
| | | 1.41 | -43.5 |
| | | 0.87 | -36.7 |
| | 20190314 | 1.14 | -44.6 |
| | | 0.15 | -5.8 |
| Taklimakan Desert | 20190409 | 0.31 | -12.3 |
| | | 0.72 | -39.5 |
| | | 0.88 | -35.4 |
| | 20190423 | 0.21 | -8.3 |
| | | 0.35 | -35.7 |
| | | 0.11 | -4.5 |
| | 20190425 | 0.79 | -44.8 |
| | | 0.98 | -52.4 |
| | | 0.55 | -29.1 |



According to the definition, the DRFE$_{dust}$ represents the DRE$_{dust}$ of a certain AOD at per unit area
during these storms in the desert dust source regions. Therefore, the DRFE$_{dust}$ can be estimated by
fitting the DRE$_{dust}$ and the AOD.

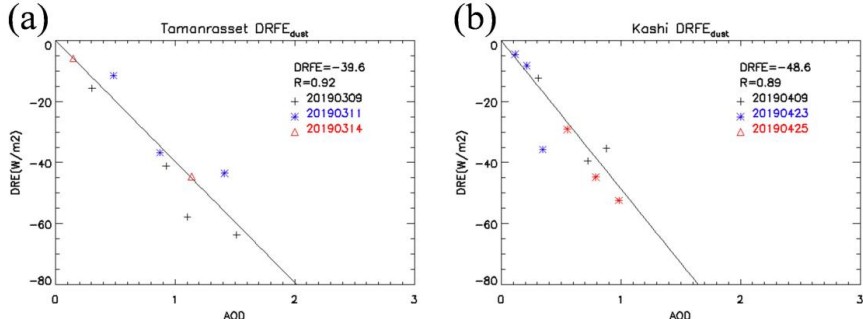

**Figure 8: DRE$_{dust}$ in (a) March 2019 over Tamanrasset and (b) April 2019 over Kashi.**
The linear relationship between the DRE$_{dust}$ and the AOD can be found during dust storms around
Tamanrasset and Kashi, which is also investigated in previous studies (Kumar et al. 2015; Jose et al.
2016). Then, the DRFE$_{dust}$ can be estimated by regressing the DRE$_{dust}$ and the AOD. In Fig. 8, the mean
DRFE$_{dust}$ of the dust storms is −39.6 Wm$^{-2}$τ$^{-1}$ over Tamanrasset and −48.6 Wm$^{-2}$τ$^{-1}$ over Tamanrasset.
The correlation coefficients are high with R = 0.92 in March 2019 over Tamanrasset and R = 0.89 in
April 2019 over Kashi. The AOD and DRE$_{dust}$ values are well correlated. Positive dust AOD is
associated with negative DRE$_{dust}$.
The equi-albedo method directly estimates the DRE$_{dust}$ and the DRFE$_{dust}$ based on the satellite
observations. Therefore, the accuracy of the results (DRE$_{dust}$ and DRFE$_{dust}$) derived from the
equi-albedo method is highly dependent on the accuracy of satellite observations. Therefore, the
uncertainties of the DRFE$_{dust}$ derived from the equi-albedo method mainly include the instantaneous
SW flux error from CERES measurements, the estimation uncertainties of the F$_{clr}$ over the dust storm
region, and the uncertainty in the deep blue AOD product. Beside that, according to our sensitivity test
in the previous studies (Tian et al., 2019), the atmospheric profile, water vapor and height of dust layer
have insignificant influence on SW radiative flux at the TOA. It is reasonable to use same water vapor
and pre-defined vertical distribution for dust aerosols in one scene of satellite data. However, the
assumption of pixels has same water vapor and pre-defined aerosol vertical distribution over one scene
of satellite data still cause small uncertainty.

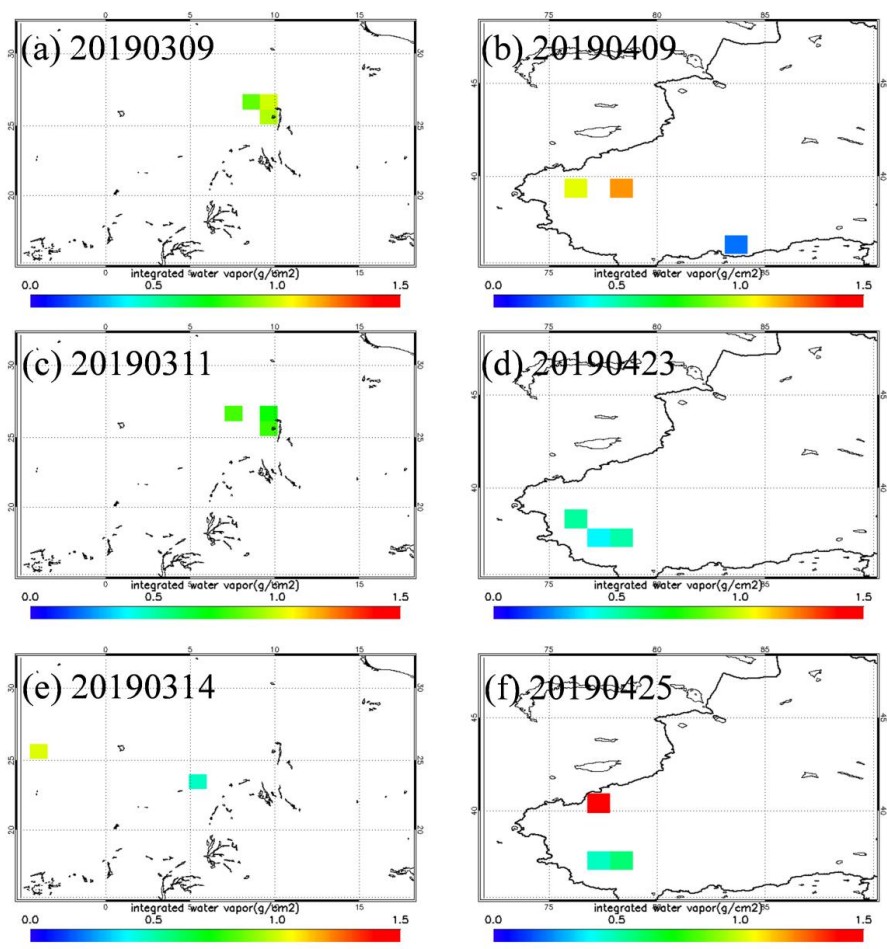

**Figure 9: Integrated water vapor (g/cm$^2$) from European Centre for Medium-range Weather Forecasts (ECMWF) reanalyses dataset on March 2019 over Tamanrasset and on April 2019 over Kashi.**

Fig.9 shows integrated water vapor from ECMWF reanalyses dataset on March 2019 over Tamanrasset and on April 2019 over Kashi. The integrated water vapor varies little over research areas, the regional mean differences are 0.51g/cm$^2$ and 0.18g/cm$^2$ over Kashi and Tamanrasset, respectively. In order to test the uncertainty caused by the varies of integrated water vapor over research areas, we calculated SW radiative flux at the TOA in difference of integrated water vapor based on SBDART model.



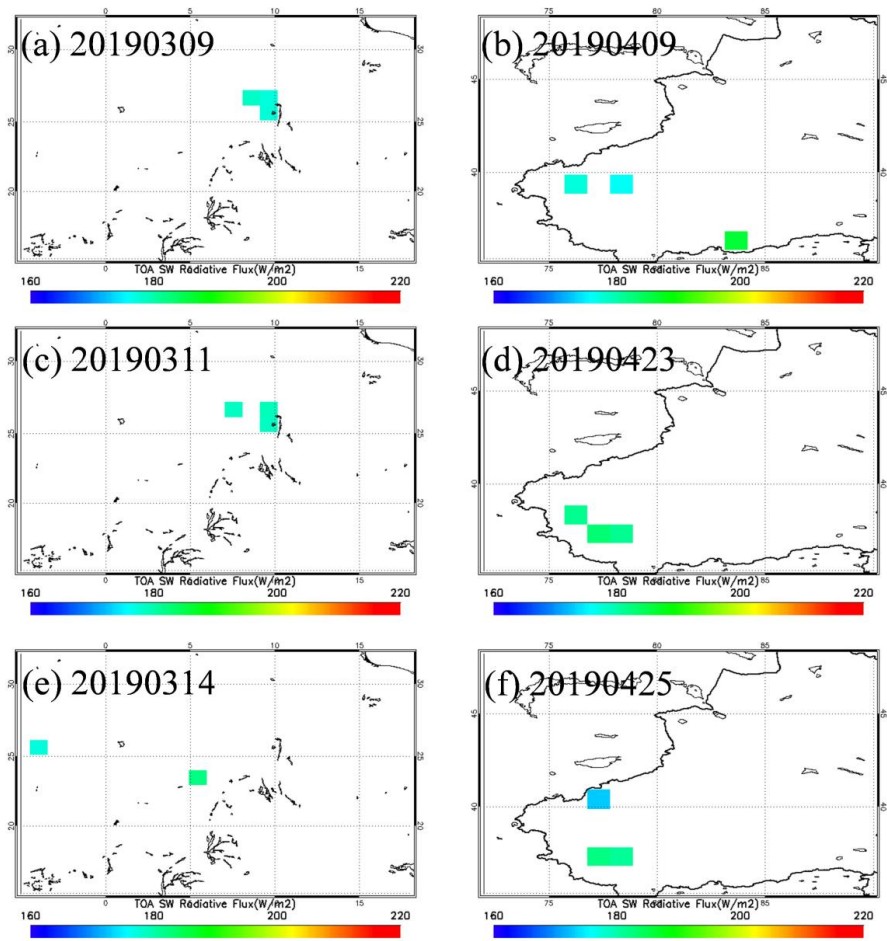


**Figure 10:   SBDART simulated clear-sky TOA radiative flux using integrated water vapor (g/cm2) from**

**ECMWF reanalyses dataset on March 2019 over Tamanrasset and on April 2019 over Kashi.**

Fig.10 shows SBDART simulated clear-sky TOA radiative flux using integrated water vapor from
ECMWF reanalyses dataset on March 2019 over Tamanrasset and on April 2019 over Kashi. The
regional mean differences of TOA radiative flux are 2.21% and 0.85% over Kashi and Tamanrasset,
respectively. This indicates the varies of integrated water vapor caused uncertainties of TOA radiative
flux are 2.21% and 0.85% over Kashi and Tamanrasset.
For the assumption of vertical profile for dust aerosols, we also tested the sensitivity of radiative
flux at the top of atmosphere to changes height of dust layer with SBDART model.



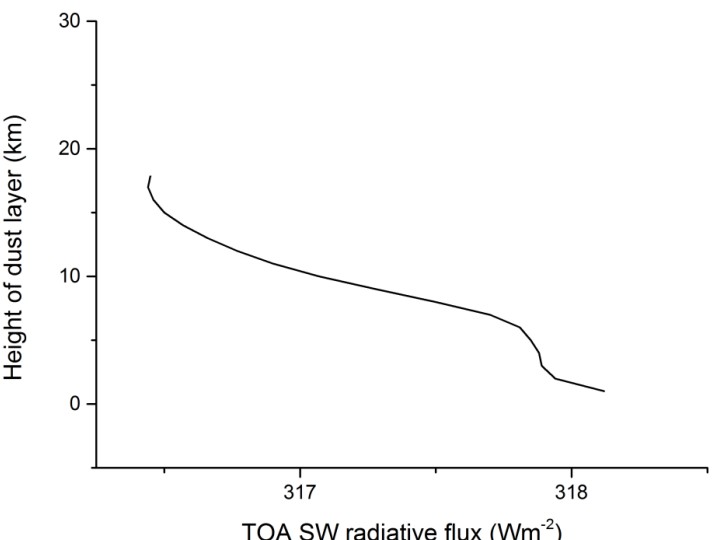


**Figure 11: The sensitivity test of SW radiative flux at the TOA to changes height of dust layer.**

As Fig.11 shown, the SW radiative flux at the TOA was decreased with the height of dust layer
was increased from 0km to 18km. However, the contents of the SW radiative flux change little with the
height of dust layer increased (within 1.5Wm$^{-2}$, 0.47%), which is little than CERES observation errors.
According to our previous study (Tian et al., 2019), the instantaneous SW flux error from CERES
measurements is about 3.13%, the estimation uncertainty of the $F_{clr}$ is 3.15%, the uncertainty of the
deep blue AOD retrieved by MODIS is about 15% (Sayer et al., 2014), and the uncertainties of using
same water vapor (2.21% and 0.85% over Kashi and Tamanrasset, respectively) and pre-defined
aerosol vertical distribution (0.47%) over one scene of satellite data. Then, the total uncertainties of the
DRFE$_{dust}$  can be calculated by the equation Eq. (3) (Zhang et al., 2005).
$U_t = \exp[\sum(logU_s)^2]^{1/2}$                                                                                                             (3).
$U_s$ is the synthetical uncertainty factor of each source of the uncertainty (including the
instantaneous SW flux error from CERES measurements, the estimation uncertainty of the $F_{clr}$, and the
uncertainty of the deep blue AOD retrieved by MODIS). $U_t$ is the total uncertainty of the  DRFE$_{dust}$,
which is 25.37% and 28.19% (10.0 Wm$^{-2}$τ$^{-1}$ and 13.7 Wm$^{-2}$τ$^{-1}$) in Tamanrasset and Kashi, respectively.
Therefore, the DRFE$_{dust}$ are $-39.6 \pm 10.0$ Wm$^{-2}$τ$^{-1}$ in March 2019 over Tamanrasset and $-48.6 \pm 13.7$
Wm$^{-2}$τ$^{-1}$ in April 2019 over Kashi.
**4 Deriving DRFE$_{dust}$ from the RTM simulations**
**4.1 Dust microphysical properties**
The focuses of this paper are the differences in the dust microphysical properties from different
dust source regions and the impacts of the dust microphysical properties on the DRFE$_{dust}$ simulation. As
important parameters concerning the radiative impacts, the volume size distribution and the refractive
index of the dust aerosol are compared in the dust storms over Tamanrasset and Kashi detected by
MODIS.

(a) Real parts of the complex refractive index over Sahara Desert

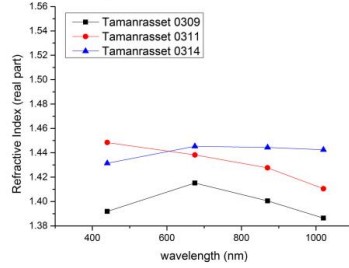

(b) Real parts of the complex refractive index over Taklimakan Desert

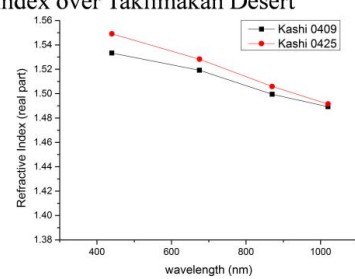

(c) Imaginary parts of the complex refractive index over Sahara Desert

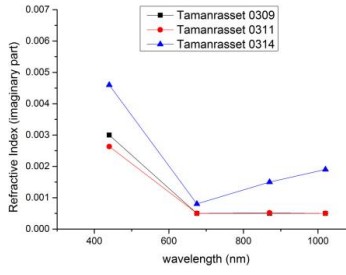

(d) Imaginary parts of the complex refractive index over Sahara Desert

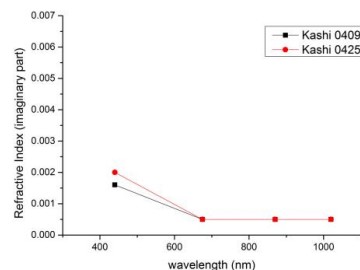


**Figure 12: Real and imaginary parts of the dust complex refractive index from the Sahara Desert and the**
**Taklimakan Desert.**
The refractive index is a measurement of the aerosol refraction and absorption efficiency. Aerosols
with high real parts of the complex refractive index values are indicated to be scattering types.
Conversely, aerosols with high imaginary parts are indicated to be absorbing types (Zhang et al., 2006).
Figure 12 shows the real and imaginary parts of the dust complex refractive index from the Taklimakan
Desert and the Sahara Desert during the dust storms. In Fig. 12, dust aerosols from the Taklimakan
desert (Fig. 12 (b)) have higher real parts and lower imaginary parts (Fig. 12 (d)) than the Sahara desert





(Fig. 12 (a) and Fig. 12 (c)), showing that the dust aerosols from the Taklimakan Desert have stronger
scattering effects.

The volume size distribution of dust aerosols clearly shows the particle size difference between

dusty and clear-sky days. The moderate and coarse aerosol particles with a radius within 0.5–10 μm
show a significant increase under dusty conditions than those under non-dusty days. It is indicated that
the quantity of the coarse mineral dust particles increases because of the dust storms. Figure 13
illustrates the variation of the dust aerosol size distribution during the dust storms in March 2019 over
Tamanrasset and in April 2019 over Kashi. Most maximum dust aerosol size distribution peaks at the
radius of 1.71 in Tamanrasset and 2.24 in Kashi. Moreover, the peak values are higher in Kashi. It is
indicated that the dust storm is stronger in April 2019 over Kashi and the coarse mode aerosol particles
increase in particle volume compared with those in the dust storm in March 2019 over Tamanrasset.

(a) Dust aerosol size distribution over Sahara Desert

(b) Dust aerosol size distribution over Taklimakan Desert

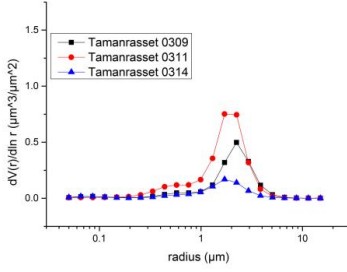

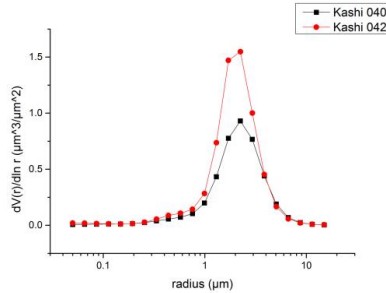


**Figure 13: Dust aerosol size distribution over (a) the Sahara Desert and (b) the Taklimakan Desert.**
**4.2 Dust optical properties**

The dust optical properties can be calculated by synergistically using the real and imaginary parts of

the dust complex refractive index and the dust aerosol size distribution.

The SSA and the ASY are two key parameters determining the $DRE_{dust}$ and the $DRFE_{dust}$. Accurate

measurements of the SSA and the ASY are important for the assessment of the direct effect of aerosols
on climate (Qie et al., 2019). The dust aerosol optical properties are calculated by using the Mie theory,
the T-matrix method, and the AERONET inversion products (Dubovik and King, 2000;Dubovik et al.,

2006).



The SSA is presented as the ratio between the aerosol scattering and extinction coefficients. The
dust SSA describes the scattering properties of the dust aerosols. The SSA can largely determine the
magnitudes and signs of the $DRE_{dust}$ and the $DRFE_{dust}$. Strongly scattering dust aerosols (i.e., SSA = 1)
always cause negative $DRE_{dust}$. By contrast, low SSA aerosols often cause positive $DRE_{dust}$, especially
over high LSA regions as the light absorbed by the aerosols can reduce the cooling effect. A high SSA is
correlated with low real parts of the complex refractive index, while a strong absorption is correlated
with a high imaginary part of the complex refractive index. Together with the size distribution, real parts
of the complex refractive index can determine the magnitude of the SSA.

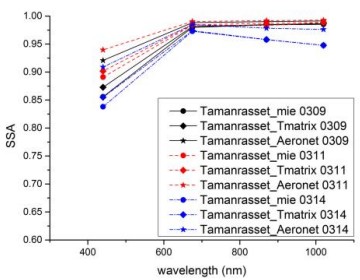

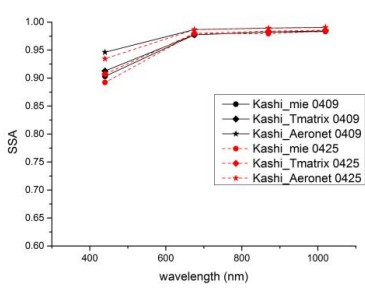


**Figure 14: Single scattering albedo from (a) the Sahara Desert and (b) the Taklimakan Desert..**
Figure 14 shows the variabilities of the dust aerosol SSA between different dust source regions and
different calculation methods. In Fig. 14, the maximum SSA value mostly occurs at the wavelength of
1020 nm, which indicates that the SSA is dependent on wavelength. Moreover, dust aerosols from the
Taklimakan desert (Kashi) in the figure have higher SSA value using both the Mie theory and the
T-matrix method. The higher value of SSA shows that dust aerosol particles scatter more
predominantly and strongly in the Taklimakan desert (Kashi), which may cause more significant
negative radiative forcing than the dust aerosols from the Sahara Desert (Tamanrasset).



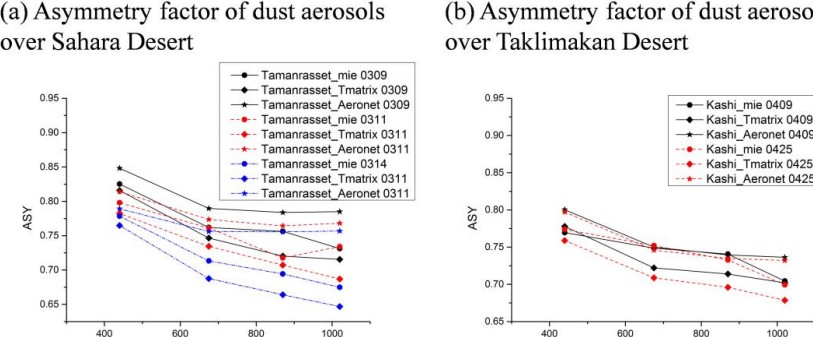

**Figure 15: Asymmetry factor in (a) the Sahara Desert and (b) the Taklimakan Desert.**

The ASY indicates the relative strength of the forward scattering, which determines the integrated

fractions of the energy that scatter backward and forward. The dust aerosol particles with sharp peaks in
the forward direction (0° scattering angle) have positive ASY. The ASY value increases with the
particle size.

The ASY in Fig. 15 shows marked spectral variation with higher values at shorter wavelengths. It

can be found that the dust aerosols from the Sahara Desert (Tamanrasset) have higher values of the ASY
than those from the Taklimakan desert (Kashi) in both the Mie theory and the T-matrix method. The
high values (over 0.80 at 440 nm) reflect the dominance of the absorbing of dust aerosols. The stronger
backward scattered energy may cause higher negative radiative forcing in the Taklimakan Desert
(Kashi).

According to the analyses of the microphysical properties and the optical properties, the dust

aerosols from the Taklimakan Desert (Kashi) scatter strongly. The negative DRFE$_{dust}$ from the
Taklimakan desert (Kashi) should be more significant than those from the Sahara Desert (Tamanrasset).
The results are in good agreement with those estimated by the satellite observations.



**4.3 DRFE$_{dust}$ derived from the RTM simulations**

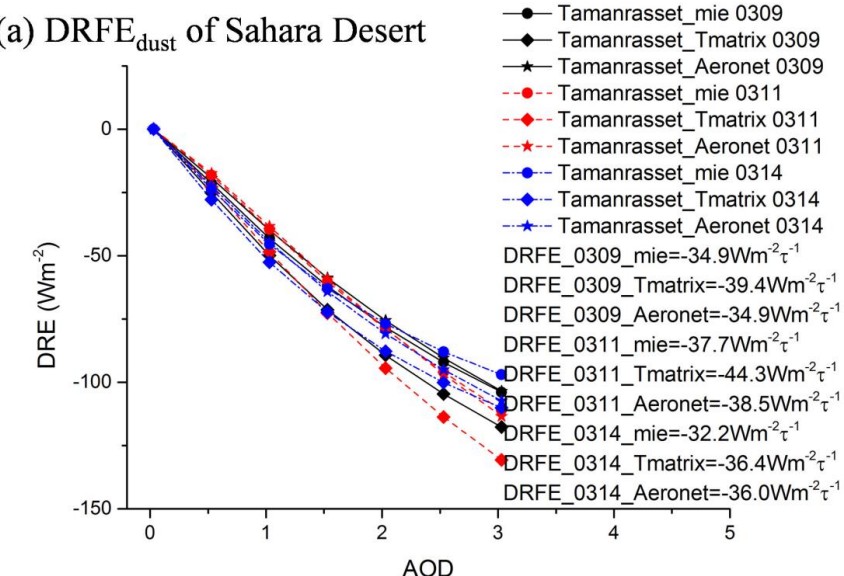

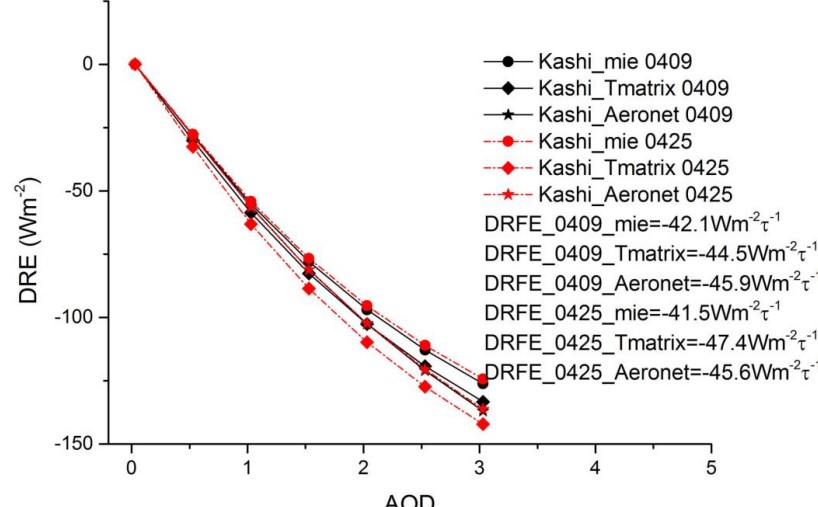


**Figure 16: DRFE$_{dust}$ simulated by the SBDART in (a) Tamanrasset and (b) Kashi.**

The DRFE$_{dust}$ estimated directly by the satellite observation is compared with that simulated by the

SBDART to verify the reliability. As shown in Fig. 16, with higher aerosol scattering (higher SSA) and



higher backward scattering coefficients (lower ASY), the negative DRFE$_{dust}$ from Kashi is more
significant. The mean DRFE$_{dust}$ from Kashi is -37.1 W m$^{-2}$ $\tau^{-1}$. The dust aerosols from Kashi have
stronger cooling effects than those from Tamanrasset, in which the mean DRFE$_{dust}$ is -44.5 W m$^{-2}$ $\tau^{-1}$.
The results are in good agreement with those estimated by the satellite observations. The DRFE$_{dust}$
estimated by the dust optical properties derived from the T-matrix method and the AERONET is closer
to those estimated by the satellite observations, which indicates that most dust aerosols are
non-spherical in the natural environment.
The results also show that the dust microphysical properties can significantly influence the
DRFE$_{dust}$. The mean difference of the DRFE$_{dust}$ between Tamanrasset and Kashi is 9.0% (7.4 W m$^{-2}$ $\tau^{-1}$).
Even for the same dust microphysical property, the DRFE$_{dust}$ varies significantly according to whether
the dust particles are considered spherical or non-spherical in different methods. For the differences of
the DRFE$_{dust}$ estimated using different methods, the mean standard deviations are 7.6% (2.8 W m$^{-2}$ $\tau^{-1}$)
in Tamanrasset and 6.8% (3.0 W m$^{-2}$ $\tau^{-1}$) in Kashi. Moreover, Li et al. (2020) pointed out that the
atmospheric profile, LSA and SZA, can also influence the simulation of the instantaneous DRFE$_{dust}$,
which agrees with our previous study (Tian et al., 2019). Additionally, it is difficult for climate models
or in-situ measurements to get the real distribution of the aerosol properties at a large spatial extent.
Also, it is hard to evaluate the uncertainties in radiative transfer simulations. It can cause significant
errors in evaluating the modulating effects of the mineral dust aerosols on climate (Huang et al.,
2009;Li et al., 2020).
**5 DRFE$_{dust}$ in the satellite-based observation and the simulation of the RTM**
According to the analyses of the dust aerosol microphysical properties and optical properties, the
dust aerosols from the Taklimakan Desert (Kashi) should scatter strongly. The RTM simulation results
are in good agreement with the results estimated by the satellite observation. Previous studies also
estimated the DRFE$_{dust}$ in the Sahara Desert and the Taklimakan Desert (Li et al., 2020;Li et al.,
2004;Garcá et al., 2012;Xia and Zong, 2009), which validate our results.
**Table 2: SW DRFE$_{dust}$ from different studies.**

| Dust source regions | Research | | | Model/Method | DRFE$_{dust}$ (Wm$^{-2}$$\tau^{-1}$) | Description |
|---|---|---|---|---|---|---|
| Sahara | Garcá | et | al | Ground+GAM | ~ -35 | AERONET |



| Desert | (2012) | E | | DRFE$_{dust}$ in December-January -February, with LSA<0.3. |
|---|---|---|---|---|
| | Li et al (2004) | Satellite+SBD ART | -35±3 (summer) -26±3 (winter) | Binned mean fitting TOA diurnal mean DRFE$_{dust}$ over the Atlantic Ocean near the African coast. |
| | This paper | Satellite Satellite+SBD ART | -39.6±10.0 (Satellite) -32.2 ~ -44.3 (SBDART) | |
| Taklimakan Desert | Li et al (2020) | Ground+SBD ART | -45~-50 | Instantaneous DRFE$_{dust}$ at 04:08 UTC. |
| | García et al (2012) | Ground+GAM E | ~ -45 | AERONET DRFE$_{dust}$ in March-April-May, with LSA<0.3. |
| | Xia and Zong (2009) | Satellite + SBDART | -48.1 | Instantaneous DRFE$_{dust}$ at about 05:00 UTC. |
| | This paper | Satellite Satellite+SBD ART | -48.6±13.7 (Satellite) -41.5 ~ -47.4 (SBDART) | |


443  Table 2 illustrates the SW DRFE$_{dust}$ of the Sahara Desert and the Taklimakan Desert in previous

444 studies. García et al. (2012) evaluated the DRFE$_{dust}$ based on the GAME model and the AERONET

445 retrievals, which indicated that the mean DRFE$_{dust}$ is around $-35$ W m$^{-2}$ $\tau^{-1}$ in the Sahara Desert and

446 $-45$ W m$^{-2}$ $\tau^{-1}$ in the Taklimakan Desert in similar observational conditions (i.e., for the SZA between

447 55 ° and 65 °, for the LSA below 0.3). Li et al. (2004) estimated the diurnal mean DRFE$_{dust}$ at the TOA

448 ($-35 \pm 3$ W m$^{-2}$ $\tau^{-1}$ in summer; $-26 \pm 3$ W m$^{-2}$ $\tau^{-1}$ in winter) over the Atlantic Ocean near the African

449 coast. The results indicated that lower uncertainties are derived from the standard deviation of the

450 best-fit curve around the observed points due to the binned mean fitting. For the Taklimakan Desert, Li

451 et al. (2020) estimated the instantaneous SW DRFE$_{dust}$ at 04:08 UTC (around $-43$ W m$^{-2}$ $\tau^{-1}$) at the

452 TOA of Kashi on 25 April 2019. In this paper, the DRFE$_{dust}$ of the Taklimakan Desert is estimated with

453 the same dust properties referring to the works of Li et al. (2020). Furthermore, Xia and Zong (2009)



used both the satellite data and the SBDART model to represent the instantaneous (about 05:00 UTC)
SW $DRFE_{dust}$, which is $-48.1$ W m$^{-2}$ $\tau^{-1}$ at the TOA(Xia and Zong, 2009). Through comparison, it is
found that the satellite-based equi-albedo method and the SBDART model-derived SW $DRFE_{dust}$ are
$-39.6 \pm 10.0$ W m$^{-2}$ $\tau^{-1}$ and $-32.2$ to $-44.3$ W m$^{-2}$ $\tau^{-1}$ at the TOA over the Sahara Desert, respectively,
which are $-48.6 \pm 13.7$ W m$^{-2}$ $\tau^{-1}$ and $-41.5$ to $-47.4$ W m$^{-2}$ $\tau^{-1}$ at the TOA over the Taklimakan Desert,
respectively. The methods and results in these studies are comparable despite the differences. The
results show that the negative $DRFE_{dust}$ from the Taklimakan Desert is more significant than those from
the Sahara Desert. As the SZA and LSA variations are considered in these studies, the results in this
paper are reasonable and reliable. The compared results show that the $DRFE_{dust}$ derived from the
satellite-based equi-albedo method is closer to that in previous studies with lower uncertainty. The
$DRFE_{dust}$ estimated by the satellite-based equi-albedo method is obtained without the dust
microphysical properties being assumed. The uncertainties are mostly caused by observation errors.
Therefore, the uncertainties can be evaluated more reasonably. It provides a direct way to validate the
$DRE_{dust}$ and the $DRFE_{dust}$.
**6 Discussion and conclusions**
This study analyzes the differences in the dust microphysical properties and the $DRFE_{dust}$ over the
Taklimakan Desert and the Sahara Desert during dust storms. The satellite-based equi-albedo method
and the RTM are both used to estimate the $DRFE_{dust}$ in this study. By comparing the results from
different methods and dust source regions, the $DRFE_{dust}$ differences caused by dust microphysical
properties and particle shapes are discussed.
The results show that the dust aerosols from the Taklimakan Desert have higher aerosol scattering
(higher SSA) and backward scattering coefficients (lower ASY), and it causes more significant
negative $DRFE_{dust}$ ($-48.6 \pm 13.7$ W m$^{-2}$ $\tau^{-1}$ by the satellite; $-41.5$ to $-47.4$ W m$^{-2}$ $\tau^{-1}$ by the SBDART)
than that in the Sahara Desert ($-39.6 \pm 10.0$ W m$^{-2}$ $\tau^{-1}$ by the satellite; $-32.2$ to $-44.3$ W m$^{-2}$ $\tau^{-1}$ by the
SBDART). It is indicated that the dust microphysical properties and particle shapes can significantly
influence on the $DRFE_{dust}$. The information on the accurate dust microphysical properties and dust
origins is highly required in the $DRFE_{dust}$ simulation. The scant measurements on dust microphysical
properties can cause large uncertainties in simulating the $DRFE_{dust}$. Previous studies proved that the





results in this paper are reasonable and reliable. The DRFE$_{dust}$ derived from the satellite-based
equi-albedo method is close to the results in previous studies.

However, there are still uncertainties in the simulation of the DRFE$_{dust}$. In contrast, the DRFE$_{dust}$

can be estimated directly from the satellite observation using the equi-albedo method without any
assumptions of the microphysical properties of dust aerosols. It has unique advantages in estimating the
DRFE$_{dust}$. Also, it can validate the DRE$_{dust}$ and the DRFE$_{dust}$ derived from the numerical models more
directly.
**Data availability**

The CERES data can be accessed from the Atmospheric Sciences Data Center of NASA Langley

Research Center (https://ceres.larc.nasa.gov/order_data.php). The AQUA/MODIS aerosol Products
(MYD04_L2) can be accessed from the NASA Level-1 and Atmosphere Archive and Distribution
System    (LAADS)    Distributed    Active    Archive    Center    (DAAC)    website
(https://ladsweb.modaps.eosdis.nasa.gov/). The MODIS albedo products (MCD43C3 Version 6) can be
accessed from the NASA LP DAAC website (https://lpdaac.usgs.gov/tools/data-pool/). The
AERONET data were obtained from the AERONET website (http://aeronet.gsfc.nasa.gov).
**Author contributions**

PZ and LC designed the study, and LT performed the study with suggestions from PZ and LC. LB

improved the scattering calculating method of dust particles. Both authors contributed to the writing of
this article.
**Competing interests**

The authors declare that they have no conflict of interest.

**Special issue statement**

This article is part of the special issue "Satellite and ground-based remote sensing of aerosol

optical, physical, and chemical properties over China". It is not associated with a conference.



**Acknowledgments**
We acknowledge the groups of MODIS, CERES, ECMWF, and AERONET for providing the
AOD, LSA, integrated water vapor, aerosol microphysical, and optical properties products. We also
thank the SBDART group for making SBDART available. We thank Nanjing Hurricane Translation for
reviewing the English language quality of this paper.
**Financial support**
This work was funded by the National Key R&D Program of China (grant number
2018YFB0504900 and 2018YFB0504905), National Natural Science Foundation (grant number

41675036).

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
