# Peer review of "Estimating Radiative Forcing Efficiency of Dust Aerosol"

_Atmospheric Chemistry and Physics, 2020_

## Referee Comment (RC2)

The authors estimated the direct radiative forcing efficiency of the dust aerosol ($DRFE_{dust}$) at some points in the Taklimakan Desert and the Sahara Desert by employing the satellite-based equi-albedo method and the radiative transfer model (RTM) simulation. In the manuscript, the authors asserted that the proposed equi-albedo method has unique advantage in estimating the $DRFE_{dust}$ because there is no need to consider the microphysical properties of dust. The authors also analyzed the differences in the dust microphysical properties between two in-situ sites of the Sahara Desert and the Taklimakan Desert (i.e., Tamanrasset and Kashi. respectively) and their impacts on $DRFE_{dust}$. The quantification of the dust radiative forcing efficiency is a meaningful and still a challenging task. However, the data and materials in the manuscript do not seem to be enough to support some conclusions. In my opinion, there are several places where the manuscript should conduct additional work.

**General comments:**

1. Dust aerosol is the object of this study. The authors should elaborate how to distinguish dust particles from other kinds of aerosols. The criteria are important to make sure that the estimated DRFE in this study is for dust aerosols. The authors declared that they chose the cases of dust storms in the Sahara and the Taklimakan deserts. However, the aerosol loadings were very low for most dust storm cases in Table 1 of the manuscript (with aerosol optical depth at 0.55 $\mu m$ as low as 0.11). It is not completely convincing. The authors need to provide evidence that the estimated direct radiation effect of dust aerosol ($DRE_{dust}$) and $DRFE_{dust}$ in this study were "real" for dust.

2. In this study, the authors proposed the satellite-based equi-albedo method to estimate the radiative forcing effects and efficiencies of dust aerosols. The authors also asserted that this method has unique advantage as there is no need to consider the microphysical properties of dust. This method is on the basis of an assumption—the shortwave (SW) radiative fluxes at the top of atmosphere (TOA) of the clear sky are equal over the regions with similar land surface albedo (LSA) and solar zenith angle (SZA). On one hand, the influences of atmospheric profiles (including the vertical distributions of water vapor density and ozone density) on the upward solar radiative flux at TOA are obvious. They cannot be ignored relative to the magnitude of dust radiative forcing. The atmospheric profiles can be very different for different "clear sky" conditions over the same region. So, the assumption doesn't sound reasonable. The authors should prove it. On the other hand, "similar" is an ambiguous criterion. The authors should clarify the conditions of LSA and SZA quantitatively, which make the above assumption tenable.

3. Some important parameters and conditions were not defined clearly in this paper. For example, (1) the effects of dust aerosol particles could exhibit opposite warming or cooling effects at the TOA, in the atmosphere, or at the surface. The authors did not specify the $DRFE_{dust}$ that was estimated at at which level (i.e., at the TOA, in the atmosphere, or at the surface) in this manuscript. Although the observations of satellites are conducted at the top of the atmosphere, it does not mean $DRFE_{dust}$ only can be calculated at the TOA. (2) The direction of radiative flux (upwelling or downwelling) was also not mentioned throughout the context, which is vital information for calculating of the $DRE_{dust}$ and $DRFE_{dust}$. (3) The radiative forcing efficiency is defined as the rate at which the atmosphere is forced per unit of aerosol optical depth (AOD) taken at a reference wavelength. The reference wavelength is an important information considering that AOD varies with wavelength. The 550 nm and 500 nm are generally adopted in literature. The value of $DRFE_{dust}$ will be changed by taken different wavelength as reference. However, it was also not specified in this study. (4) The "clear sky" condition is a basic condition for estimating the $DRFE_{dust}$. But it was not clearly stated in the manuscript. Normally, the clear sky is identified as the cloud-free and low aerosol loading sky condition, other than the sky without cloud and aerosol particles. It is necessary for the authors to specify the condition of clear sky in this study. (5) The wavelength range of SW should also be given. The authors used different broadband satellite produces (e.g., land surface albedo, radiative flux) in the SW range. SW $DRFE_{dust}$ from this study and some different preivous studies were also adopted for comparison. The ranges of spectral integration in SW were not exactly same for different previous studies (e.g., 0.2-4$\mu$m, 0.28-3$\mu$m, or 0.3-5$\mu$m). Therefore, the authors should also specify the ranges of shortwave for various parameters in this study, which are surely contributed to the differences of the $DRFE_{dust}$ results. (6) It is well-known that the instantaneous aerosol radiative forcing effects and efficiency change obviously over time. As I understand it, the instantaneous DRE and DRFE were estimated at the transit moment of the Aqua in this study. The time and sky conditions corresponding to the $DRE_{dust}$ and $DRFE_{dust}$ results of the Taklimakan Desert and the Sahara Desert should be illustrated. It is important for the comparison among different results of this study and with other previous studies. (7) The LSA is a key factor which influences the Earth-atmosphere radiation budget. LSA is calculated from the white-sky albedo (WSA) and black-sky albedo (BSA) weighted by the fraction of diffuse skylight radiation. However, only WSA was shown in the paper. The BSA which is as a function of incident solar direction, has never been mentioned. The authors should give more details on how to obtain the LSA and its influence on the assumption of the equi-albedo method in this study.

4. The authors declared that there are only sparse in-situ measurements in the main dust source regions. One of the obvious advantages of satellite

measurements is to obtain the continuous spatial information in a large region. However, only the results of DRE$_{dust}$ and DRFE$_{dust}$ at a few pixels in the Taklimakan Desert and the Sahara Desert were estimated and shown in this manuscript. Readers may prefer to see the results of dust radiative forcing over the whole regions obtained by the satellite-based method in this study. I suggest the authors giving more results in Figures 7,9 and10.

5. The authors asserted that "Previous studies proved that the results in this paper are reasonable and reliable". "Table 2 illustrates the SW DRFE$_{dust}$ of the Sahara Desert and the Taklimakan Desert in previous studies. García et al. (2012) evaluated the DRFE$_{dust}$ based on the GAME model and the AERONET retrievals, which indicated that the mean DRFE$_{dust}$ is around −35 W m$^{-2}$ $\tau^{-1}$ in the Sahara Desert and −45 W m$^{-2}$ $\tau^{-1}$ in the Taklimakan Desert in similar observational conditions…" (Lines 443-446). However, there was no DRFE$_{dust}$ result in the Taklimakan Desert in the referred literature. According to García et al. (2012), only three AERONET stations in Asia (i.e., Sacol, Dalanzadgad and Yulin) were adopted as mineral dust stations in this previous study. None of them located in the Taklimakan Desert (see Figure 1 in García et al.,2012). The authors need to give a reasonable explanation on the authenticity of the above data. Moreover, the authors also need to pay special attention to that the definitions of the aerosol DRE and DRFE published in the open literature might be very different (e.g., instantaneous value, daily average value, monthly average value, multi-year average value, multi-year monthly average value). Even for the same concept, the difference in statistical method can also lead to the difference in quantity of these values. For example, daily average result is estimated by taking the average of the 24 instantaneous hourly values in some studies, but by taking the average of the instantaneous values throughout the daytime in some other studies. The authors should notice the detail of each DRFE$_{dust}$ value in the literature. They were not the same in Table 1. So, it is not appropriate to direct compare the values of different results in this table.

**More specific comments:**

1. Section "1 Introduction": The authors need to introduce the research status of dust radiative forcing, especially in the Sahara Desert and the Taklimakan Desert. Numerous existing studies have done. The authors need to survey literature and summarized them.

2. Line 112: The equation of the DRE in this study is not commonly adopted. I can't find such a definition in any of the given references.

3. Line 119 "Based on the assumption, the F$_{clr}$ were estimated…": The authors need to give more details on how to calculate F$_{clr}$ based on the

satellite observations in the equi-albedo method.

4. Line 154 "The particle aspect ratio is set to 0.8.": The authors need to provide references to support this setting.

5. Line 165 "…(AERONET) inversion products": The authors need to give more details on the inversion products.

6. Line 229: The authors need to give more specific information on the CERES single scanner footprint (SSF) level 2 dataset.

7. Lines 254-255 "The TOA SW radiative flux distribution shows the highest value over cloud conditions.": From Figure 6, this description is not always true. For example, the bottom right corner in Figure 6f.

8. Line 256: Please explain "the SW albedo of the aerosols in the cloud".

9. Line 281 "According to the definition, the $DRFE_{dust}$ represents the $DRE_{dust}$ of a certain AOD at per unit area": It is not consistent with the previous definition in section 2.

10. Line 326 "However, the contents of the SW radiative flux change little…": Please explain "the contents of the SW radiative flux".

11. Subsection "4.1 Dust microphysical properties": The authors need to give more details on the retrievals of these dust microphysical properties in this subsection.

12. Line 360: Please provide the definition of moderate aerosol particle.

13. Line 365: I am surprised at the result of the peak radius (1.71 μm) of dust particles at Tamanrasset station. The authors need to double-check it and compare with the results from other literature.

14. Line 392-393 "The higher value of SSA shows that dust aerosol particles scatter more predominantly and strongly in the Taklimakan desert (Kashi)…": That cannot be clearly obtained from Figure 14.

15. Lines 399-403 "The ASY value increases with the particle size", "It can be found that the dust aerosols from the Sahara Desert (Tamanrasset) have higher values of the ASY than those from the Taklimakan desert (Kashi) …": They seem to contradict with the previous results those the peak radius of dust particles at Tamanrasset (1.71 μm) is smaller than that at Kashi (2.24 μm).

16. Lines 403-404 "The high values (over 0.80 at 440 nm) reflect the dominance of the absorbing of dust aerosols": The authors need to give an

explanation.

17. Lines 404-406 "The stronger backward scattered energy may cause higher negative radiative forcing in the Taklimakan Desert (Kashi)": It is not always true at TOA, in the atmosphere, or at BOA.

18. Lines 416, 475: I cannot find the results of backward scattering coefficients.

19. Lines 417-419: The mean DRFE$_{dust}$ results obviously disagree with those in Figure 8. How did the authors draw a conclusion that "The results are in good agreement with those estimated by the satellite observations"?

20. Lines 424, 427-428: The authors need to give more details on the calculation of 9.0%, 7.6% and 6.8%.

21. Lines 425-426 "Even for the same dust microphysical property, the DRFE$_{dust}$ varies significantly according to whether the dust particles are considered spherical or non-spherical in different methods": Particle shape or morphology is microphysical property.

22. Table 1: Please double-check the data in this table. The ranges "-32.2~-44.3" and "-41.5~-47.4" were obtained due to the SBDART radiative transfer calculation methodology employing different Mie or T-matrix models. The differences cannot be considered as the ranges of DRFE$_{dust}$ variation.

23. Lines 462-463 "The compared results show that the DRFE$_{dust}$ derived from the satellite-based equi-albedo method is closer to that in previous studies with lower uncertainty": I cannot find the evidences that the proposed satellite-based equi-albedo method with lower uncertainty.

24. Line 466 "Therefore, the uncertainties can be evaluated more reasonably.": Please explain and certify "more reasonably".

**Some technical comments:**

The authors need to read through the manuscript carefully and correct the grammatical errors. I just picked a few of them:

1. Lines 58-60: The large spatial variability of aerosols and the lack of an adequate database on their properties makes DRE$_{dust}$ and DRFE$_{dust}$ much difficult to estimated (Satheesh and Srinivasan, 2006).

2. Lines 78-80: Thus, the assessment of the SW DRFE$_{dust}$ and microphysical properties of the dust over these regions is important for evaluating regional and global climate changes.

3. Lines 156-157: Santa Barbara Disort Atmospheric Radiative Transfer (SBDART) is an RTM that calculates the plane-parallel radiative transfer of the earth-atmosphere system (Ricchiazzi et al., 1998).

4. Lines 272-273: To avoid the influence of the LSA and SZA in estimating the $DRFE_{dust}$, pixels with LSA of 0.16–0.20 and SZA of 32–38 degrees are chosen to derive the $DRFE_{dust}$.

5. Line 324: The sensitivity test of SW radiative flux at the TOA to changes height of dust layer.

6. Lines 325-326: As Fig.11 shown, the SW radiative flux at the TOA was decreased with the height of dust layer was increased from 0km to 18km.

7. Lines 326-327: However, the contents of the SW radiative flux change little with the height of dust layer increased (within $1.5Wm^{-2}$, 0.47%), which is little than CERES observation errors.

Some figures in the manuscript were hard to read:

8. Figure 2: The longitude and latitude are not shown in the figure. The regions of the two red boxes do not seem to correspond exactly to the images of MODIS Aqua.

9. Figures 3,5-10,12-15: Axis labels and legends are very small in these figures.

10. Figure 8: The units of $DRFE_{dust}$ should be added.

11. Figures 14-16: It is recommended to use "Mie" (starting with capital letter) instead of "mie" in these figures.

12. Figures 14,16: The lines in these figures are hard to distinguish.

The full names of acronyms should be given as they appear for the first time and keep case consistent in the full text:

13. Line 114: the full name of sensor CERES should be given as they appear for the first time in the full text.

14. Figure 3: the full name of "WSA" should be given.

15. Lines 194-195 and some other places in the text: "Aqua" and "AQUA" are suggested keeping consistent.

---

## Author Comment (AC3)

**Responses to the comments**

Dear Reviewer,

Thank you for carefully reviewing our manuscript. The comments and suggestions have helped us to improve the paper.

Following these comments and suggestions, we take a lot of efforts to optimize the structure of the manuscript, add the explanation and description on important parameters and conditions of our research, re-draw the most of the imaging and correct the grammatical errors in the article to meet the reviewer's suggestion, and modify the inaccurate information to avoid the ambiguity.

We response every comments sentence by sentence. Please find the comments in blue italics and our reply in black.

**General comments:**

1.	Dust aerosol is the object of this study. The authors should elaborate how to distinguish dust particles from other kinds of aerosols. The criteria are important to make sure that the estimated DRFE in this study is for dust aerosols. The authors declared that they chose the cases of dust storms in the Sahara and the Taklimakan deserts. However, the aerosol loadings were very low for most dust storm cases in Table 1 of the manuscript (with aerosol optical depth at 0.55 μm as low as 0.11). It is not completely convincing. The authors need to provide evidence that the estimated direct radiation effect of dust aerosol (DREdust) and DRFEdust in this study were "real" for dust.

**Reply:** According to the AERONET site information, the Tamanrasset site is located on the roof of the Regional Meteorological Center (Direction Miteo Regional Sud, Office National de la Miteorologie, Algeria) at Tamanrasset. This area, free of industrial activities, is in the highlands of the Algerian Sahara. Tamanrasset and Kashi sites are similar in land surface type, altitude, and climate. Kashi locates in the vicinity of the Taklimakan Desert, represents a place affected by dust aerosols transported from the Taklimakan Desert (Li et al., 2020). Thus, **the major aerosol over Tamanrasset and Kashi is dust aerosol, the anthropogenic and marine aerosols have little contribution to total aerosol optical depth (AOD)**, especially during dust storms.

Moreover, the focuses of our study are the differences in the dust microphysical properties from different dust source regions and the impacts of the dust microphysical properties, the microphysical properties derived from Aeronet inversion products in this study. Aeronet inversion microphysical properties products are microphysical properties of mixed aerosols. In order to corresponding the results from satellite observations and model simulations, we use AOD retrieved by MODIS as the AOD of dust aerosols in this study. Previous studies also using the AOD derived from direct observations to estimate $DRFE_{dust}$ during dust storms (Zhang and Christopher, 2003;Xia and Zong, 2009;Li et al., 2020;Li et al., 2004;Jose et al., 2016a;Garcá et al., 2012), we used the results of these studies validate our results in the paper. Thus, in order to **make the results comparable to these studies**, we also use AOD derived from direct observations to estimate $DRFE_{dust}$.

We have added the explanation of the assume that all aerosols are dust aerosol in dust storms. It has been written as "Since Sahara desert and Taklimakan desert free of industrial activities, the major aerosol over Sahara desert and Taklimakan desert is dust aerosol, the anthropogenic and marine aerosols have little contribution to total AOD, especially during dust storms. Thus, we directly use the AOD retrieved by MODIS to estimate $DRFE_{dust}$ during dust storms in this study.". Please see Lines 253-256 in Page 11-12 of the revised manuscript.

[Figure]

**Figure 1: AOD at 0.55 µm ($\tau_{550}$) of the dust storm in March 2019 over Sahara desert and that in April 2019 over Taklimakan desert.**

For the dust loading, MODIS L2 deep blue AOD product of the dust storm in March 2019 over Tamanrasset and that in April 2019 over Kashi are shown in Fig. 1. The missing data are shown in white; the high dust loading regions are shown in red; the low dust loading regions are shown in blue. Fig. 1 shows that there are heavy dust storms over Sahara desert and Taklimakan desert, the dust storm regions mainly with AOD great than 1.0 detected by MODIS, the edge regions of dust storm also has relatively high AOD value (with AOD great than 0.5), and clear-sky (low dust loading) regions mainly has AOD close to 0. That is because the research regions (Sahara desert and Taklimakan desert) of this study are free of industrial activities, and far from ocean, anthropogenic and marine aerosols have little contribution to AOD. Therefore, the aerosols in this study were "real" for dust.

In previous study, we found that $DRE_{dust}$ is significantly influenced by LSA (land surface albedo) and SZA (solar zenith angle) (Tian et al., 2019). To avoid the influence of the LSA and SZA in estimating the $DRFE_{dust}$, we estimate $DRFE_{dust}$ using pixels with similar LSA and SZA.

[Figure]

**Figure 2: SW LSA and SZA over Sahara desert and Taklimakan desert derived from AQUA/MODIS.**

Fig. 2 shows the LSA and the SZA observed by the AQUA satellite over Sahara desert (Fig. 2(a1)-(a3) and Fig. 2(b1)-(b3)) and Taklimakan desert (Fig. 2(c1)-(c3) and Fig. 2(d1)-(d3)) during dust storms. The cloud-free pixels with AOD great than 0.1, the LSA of 0.16–0.20 and SZA of 32–38 degrees are chosen to derive the DRFE$_{dust}$. We can found that the regions have similar LSA and SZA with Tamanrasset and Kashi not always in the heavy dust storm regions. And **some low dust loading regions also can be used in regression to estimate the DRFE$_{dust}$.**

2.    In this study, the authors proposed the satellite-based equi-albedo method to estimate the radiative forcing effects and efficiencies of dust aerosols. The authors also asserted that this method has unique advantage as there is no need to consider the microphysical properties of dust. This method is on the basis of an assumption—the shortwave (SW) radiative fluxes at the top of atmosphere (TOA) of the clear sky are equal over the regions with similar land surface albedo (LSA) and solar zenith angle (SZA). On one hand, the influences of atmospheric profiles (including the vertical distributions of water vapor density and ozone density) on the upward solar radiative flux at TOA are obvious. They cannot be ignored relative to the magnitude of dust radiative forcing. The atmospheric profiles can be very different for different "clear sky" conditions over the same region. So, the assumption doesn't sound reasonable. The authors should prove it. On the other hand, "similar" is an ambiguous criterion.

The authors should clarify the conditions of LSA and SZA quantitatively, which make the above assumption tenable.

**Reply:** Thank you for the valuable suggestion. We tested the sensitivity of radiative flux at the top of atmosphere to changes in atmosphere profile with SBDART radiative transfer model. For the atmospheric profiles, we use the standard atmospheric profiles in the SBDART model (atmospheric profiles are listed in Table 1), and AOD=0.1, LSA=0.3, SZA=30 °used as the input parameters.

**Table 1: Shortwave flux at the top of atmosphere as Function of atmospheric profile from Radiative Transfer Calculations**

| ATMOSPHERIC PROFILE | water vapor (g/cm2) | total ozone(atm-cm) | below_10km ozone(atm-cm) | TOA radiative flux (Wm$^{-2}$) |
|---|---|---|---|---|
| TROPICAL | 4.117 | 0.253 | 0.0216 | 213 |
| MID-LATITUDE SUMMER | 2.924 | 0.324 | 0.0325 | 217 |
| MID-LATITUDE WINTER | 0.854 | 0.403 | 0.0336 | 232 |
| SUB-ARCTIC SUMMER | 2.085 | 0.350 | 0.0346 | 221 |
| SUB-ARCTIC WINTER | 0.418 | 0.486 | 0.0340 | 238 |
| US62 | 1.418 | 0.349 | 0.0252 | 227 |

Table 1 lists the shortwave flux at the top of atmosphere as function of atmospheric profile. The sensitivity test result shows that the influence of the atmospheric profile on Radiative flux at the top of atmosphere is insignificant (with difference less than 4.5%).

Therefore, **it is reasonable to assume the atmospheric profile have few influence of SW DRE in one scene of satellite data**. And the atmospheric profile variety caused Radiative flux difference will not affect $DRE_{dust}$, because the difference will be eliminate when $F_{clr}$ subtracting $F_{dust}$. The atmospheric profile, water vapor and height of dust layer have insignificant influence on $DRFE_{dust}$ at the TOA. **The uncertainties caused by the water vapor and height of dust layer also estimated in the manuscript.** Please see Lines 349-375 in Page 17-19 of the revised manuscript.

In this research we attempt to eliminate the influence of LSA, and to estimate the clear-sky shortwave radiative flux over dust storm region, based on the assumption that $F_{clr}$ is the same as the clear-sky TOA radiative flux in dust storm regions, where the LSA and SZA are similar with that in the dust storm regions ($LSA \approx LSA_{dust}$ & $SZA \approx SZA_{dust}$). Therefore, it can be considered to be in the same condition when the difference between LSA is less than 0.01, and the difference between SZA is less than 0.2 °. This keeps good consistency of $F_{clr}$ between clear and dust storm area, and ensures that enough pixels could match the condition (Tian et al., 2019).

To avoid the influence of the LSA and SZA in estimating the $DRFE_{dust}$, we estimate $DRFE_{dust}$ using pixels with similar LSA and SZA. Fig. 2 shows LSA between 0.16-0.20, and SZA between 32 °-38 °around Tamanrasset and Kashi during dust storms. Therefore, The cloud-free pixels with AOD great than 0.1, and with the LSA of 0.16–0.20 and the SZA of 32–38 degrees are chosen to estimate the $DRFE_{dust}$.

We clarify the conditions of LSA and SZA quantitatively in the revised manuscript. Please see Lines 129-133 in Page 6, and Lines 320-322 in Page 15 of the revised manuscript.

3.    Some important parameters and conditions were not defined clearly in this paper. For example, (1) the effects of dust aerosol particles could exhibit opposite warming or cooling effects at the TOA, in the atmosphere, or at the surface. The authors did not specify the DRFEdust that was estimated at at which level (i.e., at the TOA, in the atmosphere, or at the surface) in this manuscript. Although the observations of satellites are conducted at the top of the atmosphere, it does not mean DRFEdust only can be calculated at the TOA. (2) The direction of radiative flux (upwelling or downwelling) was also not mentioned throughout the context, which is vital information for calculating of the DREdust and DRFEdust. (3) The radiative forcing efficiency is defined as the rate at which the atmosphere is forced per unit of aerosol optical depth (AOD) taken at a reference wavelength. The reference wavelength is an important information considering that AOD varies with wavelength. The 550 nm and 500 nm are generally adopted in literature. The value of DRFEdust will be changed by taken different wavelength as reference. However, it was also not specified in this study. (4) The "clear sky" condition is a basic condition for estimating the DRFEdust. But it was not clearly stated in the manuscript. Normally, the clear sky is identified as the cloud-free and low aerosol loading sky condition, other than the sky without cloud and aerosol particles. It is necessary for the authors to specify the condition of clear sky in this study. (5) The wavelength range of SW should also be given. The authors used different broadband satellite produces (e.g., land surface albedo, radiative flux) in the SW range. SW DRFEdust from this study and some different preivous studies were also adopted for comparison. The ranges of spectral integration in SW were not exactly same for different previous studies (e.g., 0.2-4μm, 0.28-3μm, or 0.3-5μm). Therefore, the authors should also specify the ranges of shortwave for various parameters in this study, which are surely contributed to the differences of the DRFEdust results. (6) It is well-known that the instantaneous aerosol radiative forcing effects and efficiency change obviously over time. As I understand it, the instantaneous DRE and DRFE were estimated at the transit moment of the Aqua in this study. The time and sky conditions corresponding to the DREdust and DRFEdust results of the Taklimakan Desert and the Sahara Desert should be illustrated. It is important for the comparison among different results of this study and with other previous studies. (7) The LSA is a key factor which influences the Earth-atmosphere radiation budget. LSA is calculated from the white-sky albedo (WSA) and black-sky albedo (BSA) weighted by the fraction of diffuse skylight radiation. However, only WSA was shown in the paper. The BSA which is as a function of incident solar direction, has never been mentioned. The authors should give more details on how to obtain the LSA and its influence on the assumption of the equi-albedo method in this study.

**Reply:** Thank you for these valuable suggestions.

(1)  In this study we estimate DRFE$_{dust}$ at the top of atmosphere (TOA). Satellites can directly observe the radiation budget of the earth at the TOA (Wielicki et al., 1998;Satheesh and Ramanathan, 2000), and the remote-sensing technique for the AOD has been developed (Remer et al., 2005;Hsu et al., 2004). Therefore, it is possible to evaluate the DRFE$_{dust}$ at the TOA directly from satellite observations. Meanwhile, the DRFE$_{dust}$ at the TOA also simulated by RTM. We have added the state of the DRFE$_{dust}$ level in the manuscript. Please see Lines 91-108 in Page 4, Line 198 and Lines 202-205 in Page 8, Line 310 and Lines 316-323 in Page 15 of the revised manuscript.

(2)  The satellite-based equi-albedo method and the RTM are both used to estimate the DRFE$_{dust}$ in this study. Therefore, the upward radiative flux at the TOA were estimated in this study. We have added the state of the radiative flux direction in the manuscript. Please see Line120 in Page 5 and Line127 in Page 6 of the revised manuscript.

(3)  In this paper, the deep blue AOD (0.55μm) data are used to discriminate the dust storm regions,

and the deep blue algorithm retrieval AOD at 0.55μm. We have mentioned the wavelength of AOD in the description of MODIS products (Please see section 3.1).

(4) The TOA radiative flux in the clear-sky area is derived from CERES pixels in cloud-free and non-aerosol conditions that have AOD smaller than 0.02, as determined by the MODIS data in Christopher et al. (2002) method (Christopher and Zhang, 2002). It is too small for the deep blue algorithm to be retrieved. In the equi-albedo method we set the cloud-free pixels with AOD smaller than 0.1 as the clear-sky pixels (Tian et al., 2019). We clarify the condition of clear sky in this study. Please see Lines 129-130 in Page 6 of the revised manuscript.

(5) Thank you for your reminding. Here the instantaneous SW channel (0.3–5.0 μm) radiative flux at the TOA from CERES SSF level 2 dataset and SW channel (0.3–5.0 μm) land surface albedo product (MCD43) were used. Other studies estimated $DRFE_{dust}$ with the broadband wavelength in 0.3–5.0 μm, which consistent with our study. Garcá et al. (2012) estimated $DRFE_{dust}$ with the broadband wavelength in 0.2–4.0 μm, to keep the scientific and rigorous of the research we do not compared the $DRFE_{dust}$ derived from works of Garcá et al. (2012) with our results any longer. We have revised and clarified the the wavelength range of SW in the manuscript. Please see Line 265 and Line 273 in Page 12, Lines 496-506 in Page 27 of the revised manuscript.

(6) It is true that the instantaneous aerosol radiative forcing effects and efficiency change obviously over time. That mainly caused by the changes of SZA over time, and SZA have important influence on SW $DRE_{dust}$ and $DRFE_{dust}$. To avoid the influence of the SZA in estimating the SW $DRFE_{dust}$, we estimate $DRFE_{dust}$ using pixels with similar SZA. Therefore, the transit time of AQUA satellite would not influence on the estimation of $DRFE_{dust}$. We also added the description of the time of AQUA satellite cross over Taklimakan Desert and the Sahara Desert to meet the reviewer's suggestion. Please see Lines 235-236 in Page 10 of the revised manuscript.

[Figure]

**Figure 3: True color images and cloud detections from AQUA/MODIS observations.**

Fig. 3 shows the true color images (Fig. 4(a1)-(a3) over Sahara desert, and Fig. 4(c1)-(c3) over Taklimakan desert) and cloud detections (Fig. 4(c1)-(c3) over Sahara desert, and Fig. 4(d1)-(d3) over Taklimakan desert) from AQUA/MODIS observations. The true color images and cloud detections clearly show the sky conditions of Taklimakan Desert and the Sahara Desert during these dust storms, Tamanrasset site and Kashi site were not covered by clouds during these dust storms. Please see Figure 4 in Page 11 of the revised manuscript.

(7) The shortwave black-sky albedo (BSA) and white-sky albedo (WSA) are provided by the MODIS BRDF/Albedo Science Data Product (MCD43C3).

BSA (directional hemispherical reflectance) is defined as albedo in the absence of a diffuse component and is a function of SZA. WSA (bihemispherical reflectance) is defined as albedo in the absence of a direct component when the diffuse component is isotropic. BSA and WSA mark the extreme cases of completely direct and completely diffuse illumination. Here, the LSA is calculated by interpolating from BSA and WSA (Lewis and Barnsley, 1994;Schaaf et al., 2002). In this study we estimate $DRFE_{dust}$ in a restricted range of SZA. Thus, the LSA, BSA and WSA show good coincidence

We clarify the LSA we used in this study. Please see Lines 265-268 in Page 12 of the revised manuscript.

4.    The authors declared that there are only sparse in-situ measurements in the main dust source regions. One of the obvious advantages of satellite measurements is to obtain the continuous spatial information in a large region. However, only the results of DREdust and DRFEdust at a few pixels in the Taklimakan Desert and the Sahara Desert were estimated and shown in this manuscript. Readers may prefer to see the results of dust radiative forcing over the whole regions obtained by the satellite-based method in this study. I suggest the authors giving more results in Figures 7,9 and10.

**Reply:** This suggestion is pretty valuable. In previous study, we found that $DRE_{dust}$ is significantly influenced by LSA and SZA (Tian et al., 2019). To avoid the influence of the LSA and SZA in estimating the $DRFE_{dust}$, we estimate $DRFE_{dust}$ using pixels with similar LSA and SZA. We have re-plotted all the available data of $DRE_{dust}$, integrated water vapor and SBDART derived SW fluxes in these distribution maps, and put black borders around the chosen pixels in these figures.

[Figure]

**Figure 4: AOD and DRE$_{dust}$ of dust storms over the Sahara Desert in March 2019 and over the Taklimakan Desert in April 2019.**

Fig.4 shows that AOD and $DRE_{dust}$ plotted in the same image, the high dust aerosol loading regions show significant negative radiative forcing. It indicates that the dust aerosol loading is

negatively correlated with the DRE$_{dust}$ in these dust storm events. Thus, dust aerosols have a negative radiative effect in the SW spectrum.

[Figure]

**Figure 5: Integrated water vapor (g/cm$^2$) from European Centre for Medium-range Weather Forecasts (ECMWF) reanalysis dataset over the Sahara Desert in March 2019 and over the Taklimakan Desert in April 2019.**

Fig.8 shows the integrated water vapor from ECMWF reanalysis dataset over the Sahara Desert in March 2019 and over the Taklimakan Desert on April 2019. The grids surrounded by black border are the chosen pixels to estimate the DRFE$_{dust}$. The integrated water vapor varies little over different research areas, and the mean differences of chosen pixels are 0.51g/cm$^2$ and 0.18g/cm$^2$ over the Sahara Desert and the Taklimakan Desert, respectively.

[Figure]

**Figure 6: SBDART simulated clear-sky TOA radiative flux by using integrated water vapor (g/cm$^2$) from ECMWF reanalysis dataset over the Sahara Desert in March 2019 and over the Taklimakan Desert in April 2019.**

Fig.6 shows SBDART simulated clear-sky TOA radiative flux using integrated water vapor from ECMWF reanalyses dataset on March 2019 over Sahara desert and on April 2019 over Taklimakan desert, and the grids surrounded by black border are the chosen pixels to derive the DRFE$_{dust}$. The regional mean differences of TOA radiative flux are 2.21% and 0.85% over Sahara desert and Taklimakan desert, respectively.

We re-drawed these distribution maps, and revised in the manuscript. Please see Fig.6 in Page 14, Fig.8 in Page 17, and Fig.9 in Page 18 of the revised manuscript.

5.    The authors asserted that "Previous studies proved that the results in this paper are reasonable and reliable". "Table 2 illustrates the SW DRFEdust of the Sahara Desert and the Taklimakan Desert in

previous studies. Garc á et al. (2012) evaluated the DRFE$_{dust}$ based on the GAME model and the AERONET retrievals, which indicated that the mean DRFE$_{dust}$ is around −35 W m$^{-2}$ τ$^{-1}$ in the Sahara Desert and −45 W m$^{-2}$ τ$^{-1}$ in the Taklimakan Desert in similar observational conditions…" (Lines 443-446). However, there was no DRFEdust result in the Taklimakan Desert in the referred literature. According to Garc á et al. (2012), only three AERONET stations in Asia (i.e., Sacol, Dalanzadgad and Yulin) were adopted as mineral dust stations in this previous study. None of them located in the Taklimakan Desert (see Figure 1 in Garc á et al.,2012). The authors need to give a reasonable explanation on the authenticity of the above data. Moreover, the authors also need to pay special attention to that the definitions of the aerosol DRE and DRFE published in the open literature might be very different (e.g., instantaneous value, daily average value, monthly average value, multi-year average value, multi-year monthly average value). Even for the same concept, the difference in statistical method can also lead to the difference in quantity of these values. For example, daily average result is estimated by taking the average of the 24 instantaneous hourly values in some studies, but by taking the average of the instantaneous values throughout the daytime in some other studies. The authors should notice the detail of each DRFEdust value in the literature. They were not the same in Table 1. So, it is not appropriate to direct compare the values of different results in this table.

**Reply:** Thank you for your reminding.

[Figure]

**Figure 7: The research regions in works of Garc á et al. (2012)**

Fig.7 shows Geographical distribution of the AERONET stations used in works of Garc á et al. (2012) (Garc á et al., 2012), these stations were grouped into 14 regions. The region of R4 in Fig.7 contained the Taklimakan Desert, and we compared the DRFE$_{dust}$ at TOA over R4 (~ −45 W m$^{-2}$ τ$^{-1}$) with our results.

Following the suggestion, we checked the loaction of AERONET stations in R4, these stations (Sacol, Dalanzadgad and Yulin) are not in Taklimakan Desert. Moreover, Garc á et al. (2012) estimated DRFE$_{dust}$ with the broadband wavelength in 0.2–4.0 μm which is different from our research (0.3-5.0 μm). Thus, it is not appropriate to direct compare the values of different results. We have revised in the manuscript. Please see Lines 496-507 in Page 27 of the revised manuscript.

Other studies are comparable. We estimated the DRFE$_{dust}$ of the Taklimakan Desert with the same dust properties referring to the works of Li et al. (2020) (Li et al., 2020), Li et al. (2020) estimated the

instantaneous SW (with the broadband wavelength in 0.2–4.0 μm) $DRFE_{dust}$ around Kashi during dust storms. The $DRFE_{dust}$ derived from works of Li et al. (2020) are around -45 $Wm^{-2}\tau^{-1}$~50 $Wm^{-2}\tau^{-1}$, this can be directly compared with the result we estimated in the manuscript, and the results are in good agreement with those estimated by the satellite observations in our study.

Furthermore, Li et al. (2004) (Li et al., 2004) and Xia and Zong (2009) (Xia and Zong, 2009) also used the satellite data (with the broadband wavelength in 0.3–5.0 μm) to estimate the instantaneous SW $DRFE_{dust}$ at TOA.

**More specific comments:**

1. Section "1 Introduction": The authors need to introduce the research status of dust radiative forcing, especially in the Sahara Desert and the Taklimakan Desert. Numerous existing studies have done. The authors need to survey literature and summarized them.

**Reply:** Thanks for the valuable suggestion. Many studies have estimated the $DRFE_{dust}$ in the Sahara Desert and the Taklimakan Desert (Xia and Zong, 2009;Li et al., 2020;Li et al., 2004;Garcá et al., 2012) in different methods and conditions. We have summarized and cited these works in Section 1. Please see Lines 78-90 in Page 3-4 of the revised manuscript.

2. Line 112: The equation of the DRE in this study is not commonly adopted. I can't find such a definition in any of the given references.

**Reply:** The term "radiative forcing" has been employed in the IPCC (Intergovernmental Panel on Climate Change) Assessments to denote an externally imposed perturbation in the radiative energy budget of the Earth's climate system (Ramaswamy, 2006). Such a perturbation can be brought about by secular changes in the concentrations of radiatively active species (e.g., CO2, aerosols), changes in the solar irradiance incident upon the planet. This imbalance in the radiation budget has the potential to lead to changes in climate parameters and thus result in a new equilibrium state of the climate system.

According to the definition of direct radiative effect (DRE) and direct radiative forcing (DRF), the DRE is the instantaneous radiative impact, and the DRF is the change in DRE from preindustrial times to the present day (Heald et al., 2014).

The instantaneous DRF at the top of the atmosphere (TOA) is defined as the difference between clear ($F_{clr}$) and aerosol ($F_{aer}$) fluxes in previous studies ($DRF_{aer} = F_{clr} - F_{aer}$) (Lin et al., 2009;Li et al., 2020;Jose et al., 2016b;Christopher et al., 2006;Christopher and Zhang, 2002;Christopher et al., 2000). **To keep the description scientific and rigorous, we use DRE instead of DRF** ($DRE_{aer} = F_{clr} - F_{aer}$) in this study.

3. Line 119 "Based on the assumption, the Fclr were estimated…": The authors need to give more details on how to calculate Fclr based on the satellite observations in the equi-albedo method.

**Reply:** Thanks for the suggestion. In this research we attempt to eliminate the influence of LSA, and to estimate the clear-sky shortwave radiative flux over dust storm region, based on the assumption that $F_{clr}$ is the same as the clear-sky TOA radiative flux in dust storm regions, where the LSA and SZA are similar with that in the dust storm regions ($LSA \approx LSA_{dust}$ & $SZA \approx SZA_{dust}$).

Based on the assumption, we attempt to estimate clear-sky shortwave radiative flux over dust

storm regions, and the steps are as following:

1) TOA radiative flux in the clear-sky area is derived from CERES pixels in cloud-free and non-aerosol conditions that have AOD smaller than 0.02, as determined by the MODIS data in Christopher et al. (2002) method (Christopher and Zhang, 2002). It is too small for the deep blue algorithm to be retrieved. Here we set the pixels with AOD smaller than 0.1 as the clear-sky pixels.

2) Then, in order to eliminate the influence of LSA, $F_{clr}$ for CERES pixels that have similar LSA and SZA to the dust storm pixel is determined. Different types of land surface may have the similar LSA, we use land cover type data to ensure that the dust-free region and the dust-laden region have similar broadband integrated albedo over same land type.

[Figure]

**Figure 8: Dust storm viewed by MODIS Aqua on April 24, 2010. The red square area represents the dust storm pixel, and the blue square area represents the pixel that have similar $F_{clr}$ with the red square area, based on our assumption. Colored dots in the squares represent the mean AOD (550nm AOD retrieved by MODIS) over square areas respectively. The color of each dot represents the mean AOD value.**

The dust storm occurred on April 24, 2010 can be seen visually by Aqua MODIS (Figure 6). Following the above steps, contemporaneous surface albedo and SZA data have been used to monitor the dust storm area (the red square area in Figure 6, the mean AOD over this area is 0.05) and its match-up clear-sky area (the blue square area in Figure 6, the mean AOD over this area is 2.33) where the TOA radiative flux is similar to $F_{clr}$ over duststorm area. According to this method, the influence of land surface albedo is basically eliminated, and $F_{clr}$ over duststorm area is obtained at each pixel. We add the description on the conditions of the assumption. Please see Lines 129-133 in Page 6 of the revised manuscript.

4.   Line 154 "The particle aspect ratio is set to 0.8.": The authors need to provide references to support this setting.

**Reply:** In this study, the RTM results were used to theoretical verification of DRFE$_{dust}$ derived from satellite observation. The dust particles are assumed to sphere (aspect ratio equal to 1) and ellipsoid (aspect ratio equal to 0.8) for dust aerosol optical properties calculating. The results shows the dust aerosol optical properties were difference in particles shape assuming, and ellipsoid (aspect ratio equal to 0.8) results is closer to those estimated by AERONET inversion aerosol optical products and the satellite observations, that is indicates most dust aerosols are non-spherical in the natural environment. There may have better assumptions of aerosol particle shapes to calculate the aerosol optical properties closer to the real values, and that need further works on observation and research. In this study, we use sphere (aspect ratio set as 1.0) and ellipsoid (aspect ratio set as 0.8) to discuss the aerosol optical properties were difference in particles shape assuming.

We have added the detailed information about the particle aspect ratio setting in the manuscript. Please see Lines 168-175 in Page 7 of the revised manuscript.

5. Line 165 "…(AERONET) inversion products": The authors need to give more details on the inversion products.

**Reply:** It has been rewritten as "AERONET retrieves the physical properties of aerosols including volume size distribution and the complex refractive index, and optical properties including the SSA and the ASY (Dubovik and King, 2000;Dubovik et al., 2006)" are used as the input parameters for the SBDART model in simulating the DRFE$_{dust}$.

We have added detailed information on the inversion products. Please see Lines 187-189 in Page 8 of the revised manuscript.

6. Line 229: The authors need to give more specific information on the CERES single scanner footprint (SSF) level 2 dataset.

**Reply:** The CERES Single Scanner Footprint (SSF) Level 2 instantaneous SW Flux data available at http://ceres.larc.nasa.gov/. The CERES SSF is a unique product for studying the role of clouds, aerosols and radiation in climate. CERES TOA Flux from SSF product combines measurements with scene information from MODIS Aqua (in our study). This data corresponds to the footprints that are co-located in latitude and longitude with reference to the MODIS scene.

The radiative fluxes at TOA are derived from the CERES radiance measurements in three broad-band channel, using empirical Angular Distribution Models (ADMs). The ADMs are a function of varying scene types, such as land, ocean, cloud cover, aerosols, etc. Research shows that the uncertainty of TOA instantaneous shortwave flux is about 1.6% (4.5Wm$^{-2}$) over clear-sky land, and about 2.7% (8.4Wm$^{-2}$) over land under all-sky conditions (Su et al., 2015). Thus, TOA radiative fluxes derived from the CERES measuring radiance are reliable which could be used to estimate the DRE$_{dust}$ and the DRFE$_{dust}$ at the TOA. The DRE$_{dust}$ and the DRFE$_{dust}$ at the TOA are estimated by synergistically using MODIS and CERES products.

We have added the detailed information about the CERES SSF L2 dataset in the manuscript. Please see Lines 269-273 in Page 12 of the revised manuscript.

7. Lines 254-255 "The TOA SW radiative flux distribution shows the highest value over cloud conditions.": From Figure 6, this description is not always true. For example, the bottom right corner in Figure 6f.

**Reply:** Thank you for carefully reviewing our manuscript, it is ture that the TOA SW radiative flux not

have extremely high value in the bottom right corner in Figure 6f. That maight be because not all of the pixels are covered by cloud in the bottom right corner in Figure 6f, some of pixels are in clear-sky conditions (not covered by clouds). Thus, it is reasonable that the mixed pixel not have extremely high value of TOA SW radiative flux as cloudy pixels, and the mixed pixel also have higher TOA SW radiative flux values than dusty and clear-sky pixels. Therefore, we consider the description of "The TOA SW radiative flux distribution shows the highest value over cloud conditions." is ture for Fig.6.

8. Line 256: Please explain "the SW albedo of the aerosols in the cloud".

**Reply:** Thank you for helping us to check out the mistake. It has been rewritten as "It is due to the SW albedo of the dust aerosols and cloud are higher than land surface albedo.". Please see Line 292-293 in Page 13 of the revised manuscript.

9. Line 281 "According to the definition, the $DRFE_{dust}$ represents the $DRE_{dust}$ of a certain AOD at per unit area": It is not consistent with the previous definition in section 2.

**Reply:** The $DRFE_{dust}$ represents the $DRE_{dust}$ of per unit aerosol optical depth (AOD), which means the efficiency of the dust aerosol that affects the net radiative flux of solar radiation. We have revised this sentence in the manuscript. It has been rewritten as "According to the definition, the $DRFE_{dust}$ represents the $DRE_{dust}$ of per unit AOD during these storms in the dust source regions.". Please see Lines 329-330 in Page 16 of the revised manuscript.

10. Line 326 "However, the contents of the SW radiative flux change little…": Please explain "the contents of the SW radiative flux".

**Reply:** It has been rewritten as "the values of the SW radiative flux changed little while the height of dust layer increased". Please see Lines 373-374 in Page 19 of the revised manuscript.

11. Subsection "4.1 Dust microphysical properties": The authors need to give more details on the retrievals of these dust microphysical properties in this subsection.

**Reply:** The inversion of sun-photometry optical data to obtain particle microphysical properties has been done through numerous approaches. Currently, the AERONET inversion algorithm makes use of direct sun and sky radiance measurements (Dubovik et al., 2006;Dubovik and King, 2000).

We have added details on the retrievals of these dust microphysical properties in Section 4.1. Please see Lines 397-399 in Page 21 of the revised manuscript.

12. Line 360: Please provide the definition of moderate aerosol particle.

**Reply:** This sentence means coarse mode dust aerosol particles increase under dusty conditions. Sorry for the ambiguity sentence. We have removed the ambiguity sentence from the manuscript. Please see Lines 416-421 in Page 22 of the revised manuscript.

13. Line 365: I am surprised at the result of the peak radius (1.71 μm) of dust particles at Tamanrasset station. The authors need to double-check it and compare with the results from other literature.

**Reply:** Following the suggestion, we double-checked the aerosol size distribution in the Sahara Desert during dust storms. The results shows the aerosol size distribution peaks at the radius of 1.71μm in Tamanrasset during dust storms occurred on March 9 and 11, 2019.

We also survey previous study on the aerosol size distribution in the Sahara Desert. The works of

Guirado et al. (2014) (Guirado et al., 2014) shows the aerosol particle effective radius mainly around 1.7µm in Tamanrasset.

Table 5. Monthly means of volume particle concentration (VolCon) of total, fine and coarse modes, fine mode volume fraction ($V_f/V_t$), and effective radius ($R_{eff}$) for the period October 2006–February 2009 at Tamanrasset[a].

| Month | VolCon (µm³ µm⁻²) | | | $V_f/V_t$[b] | $R_{eff}$ (µm) | | | No. of days |
|---|---|---|---|---|---|---|---|---|
| | Total | Fine | Coarse | | Total | Fine | Coarse | |
| January | 0.04 | 0.005 | 0.04 | 0.21 | 0.63 | 0.163 | 1.92 | 38 |
| | (0.09) | (0.003) | (0.08) | (0.09) | (0.22) | (0.023) | (0.38) | |
| February | 0.06 | 0.008 | 0.05 | 0.17 | 0.70 | 0.151 | 1.89 | 27 |
| | (0.09) | (0.008) | (0.09) | (009) | (0.20) | (0.026) | (0.21) | |
| March | 0.17 | 0.015 | 0.15 | 0.11 | 0.78 | 0.141 | 1.86 | 22 |
| | (0.18) | (0.016) | (0.17) | (0.05) | (0.16) | (0.025) | (0.32) | |
| April | 0.16 | 0.014 | 0.14 | 0.11 | 0.77 | 0.145 | 1.66 | 21 |
| | (0.22) | (0.012) | (0.20) | (0.04) | (0.11) | (0.020) | (0.14) | |
| May | 0.23 | 0.017 | 0.21 | 0.09 | 0.86 | 0.133 | 1.72 | 24 |
| | (0.18) | (0.011) | (0.17) | (0.02) | (0.11) | (0.012) | (0.09) | |
| June | 0.25 | 0.019 | 0.23 | 0.10 | 0.80 | 0.129 | 1.68 | 35 |
| | (0.22) | (0.008) | (0.22) | (0.04) | (0.19) | (0.016) | (0.13) | |
| July | 0.27 | 0.025 | 0.25 | 0.13 | 0.69 | 0.122 | 1.72 | 35 |
| | (0.31) | (0.014) | (0.30) | (0.06) | (0.22) | (0.014) | (0.11) | |
| August | 0.19 | 0.022 | 0.17 | 0.16 | 0.61 | 0.123 | 1.72 | 45 |
| | (0.16) | (0.011) | (0.15) | (0.08) | (0.17) | (0.018) | (0.14) | |
| September | 0.20 | 0.018 | 0.18 | 0.10 | 0.79 | 0.139 | 1.62 | 23 |
| | (0.11) | (0.009) | (0.10) | (0.02) | (0.11) | (0.019) | (0.09) | |
| October | 0.12 | 0.014 | 0.11 | 0.13 | 0.71 | 0.143 | 1.62 | 45 |
| | (0.10) | (0.009) | (0.10) | (0.04) | (0.11) | (0.019) | (0.14) | |
| November | 0.05 | 0.008 | 0.04 | 0.19 | 0.58 | 0.146 | 1.77 | 54 |
| | (0.03) | (0.005) | (0.03) | (0.07) | (0.13) | (0.024) | (0.27) | |
| December | 0.04 | 0.007 | 0.03 | 0.22 | 0.59 | 0.159 | 1.82 | 38 |
| | (0.03) | (0.004) | (0.02) | (0.09) | (0.18) | (0.030) | (0.25) | |

[a]Corresponding standard deviations are shown in brackets.
[b]Dimensionless.

14. Line 392-393 "The higher value of SSA shows that dust aerosol particles scatter more predominantly and strongly in the Taklimakan desert (Kashi)…": That cannot be clearly obtained from Figure 14.

**Reply:** We re-drawed the image.

(a) Single Scattering Albedo of dust aerosols over Sahara Desert

[Figure]

(b) Single Scattering Albedo of dust aerosols over Taklimakan Desert

[Figure]

**Figure 9: Single scattering albedo from (a) the Sahara Desert and (b) the Taklimakan Desert.**

Figure 9 shows the variabilities of the dust aerosol SSA between different dust source regions and different calculation methods. The SSA of dust aerosol in Taklimakan desert are higher than dust aerosol in Sahara desert in most dust storms. Please see Fig.13 in Page 23 of the revised manuscript.

15. Lines 399-403 "The ASY value increases with the particle size", "It can be found that the dust

aerosols from the Sahara Desert (Tamanrasset) have higher values of the ASY than those from the Taklimakan desert (Kashi) …": They seem to contradict with the previous results those the peak radius of dust particles at Tamanrasset (1.71 μm) is smaller than that at Kashi (2.24 μm).

**Reply:** It is ture that larger dust particles usually have higher values of ASY. However, the ASY here is the mean ASY of all the dust particles during dust storms, not the ASY of dust particles in peak radius. Therefore, it is reasonable that the mean ASY of dust aerosols in Tamanrasset is higher than the dust aerosols in Kashi, although the peak radius of dust particles at Tamanrasset (1.71 μm) is smaller than that at Kashi (2.24 μm).

Moreover, the ASY derived from AERONET (the pentagrams in Fig. 14 of the revised manuscript) also shows the mean ASY of dust aerosols in Tamanrasset is higher than the dust aerosols in Kashi. This proves the ASY calculted by Mie and T-matrix methods are reliable, and it is reasonable that the mean ASY of dust aerosols in Tamanrasset is higher than the dust aerosols in Kashi during the dust storms.

16.  Lines 403-404 "The high values (over 0.80 at 440 nm) reflect the dominance of the absorbing of dust aerosols": The authors need to give an explanation.

**Reply:** Sorry for the ambiguity sentence. We have removed this sentence from the manuscript. Please see Lines 449-450 in Page 23 of the revised manuscript.

17.  Lines 404-406 "The stronger backward scattered energy may cause higher negative radiative forcing in the Taklimakan Desert (Kashi)": It is not always true at TOA, in the atmosphere, or at BOA.

**Reply:** In this study we evaluate the $DRFE_{dust}$ at TOA both using satellite observations and RTM simulations. The stronger backward scattered energy may cause higher negative radiative forcing at TOA. Therefore, it been rewritten as "The stronger backward scattered energy may cause higher negative radiative forcing in the Taklimakan Desert (Kashi) at the TOA.".

Please see Line 457 in Page 24 of the revised manuscript.

18.  Lines 416, 475: I cannot find the results of backward scattering coefficients.

**Reply:** Sorry for the ambiguity sentence. It has been rewritten as "As shown in Fig. 15, with higher aerosol scattering (higher SSA in Fig. 13) and higher backward scattering (lower ASY in Fig. 14), the negative $DRFE_{dust}$ from Kashi is more significant.". Please see Line 467 in Page 25 of the revised manuscript.

19.  Lines 417-419: The mean $DRFE_{dust}$ results obviously disagree with those in Figure 8. How did the authors draw a conclusion that "The results are in good agreement with those estimated by the satellite observations"?

**Reply:**

[Figure]

**Figure 10: DRE$_{dust}$ derived from satellite observations in (a) March 2019 over Tamanrasset and (b) April 2019 over Kashi.**

In Fig. 10, the mean DRFE$_{dust}$ of the dust storms is $-39.6$ Wm$^{-2}$τ$^{-1}$ over Tamanrasset and $-48.6$ Wm$^{-2}$τ$^{-1}$ over Tamanrasset.

[Figure]

[Figure]

**Figure 11: DRFE$_{dust}$ simulated by the SBDART in (a) Tamanrasset and (b) Kashi.**

The DRFE$_{dust}$ estimated directly by the satellite observation is compared with that simulated by the SBDART to verify the reliability. As shown in Fig. 11, the negative DRFE$_{dust}$ from Kashi is more significant. The mean DRFE$_{dust}$ in Kashi is $-44.5 Wm^{-2}\tau^{-1}$, and mean DRFE$_{dust}$ in Tamanrasset is $-37.1 Wm^{-2} \tau^{-1}$. The results are in good agreement with those estimated by the satellite observations.

20. Lines 424, 427-428: The authors need to give more details on the calculation of 9.0%, 7.6% and 6.8%.

**Reply:** Thank you for helping us to check out the mistake. The mean difference of the $DRFE_{dust}$ between Tamanrasset and Kashi (7.4 W $m^{-2}$ $\tau^{-1}$, 18.14%) is derived from difference between the mean $DRFE_{dust}$ in Tamanrasset (-37.1 W $m^{-2}$ $\tau^{-1}$) and in Kashi (-44.5 W $m^{-2}$ $\tau^{-1}$); the mean standard deviations of the $DRFE_{dust}$ estimated using different method in Tamanrasset (2.8 W $m^{-2}$ $\tau^{-1}$, 7.6%) and in Kashi (3.0 W $m^{-2}$ $\tau^{-1}$, 6.8%) are derived from the mean standard deviations of the $DRFE_{dust}$ results which using same dust aerosol properties and in different methods. We have added the calculation details and corrected the mistake in the revised manuscript. Please see Line 470-475 in Page 26 of the revised manuscript.

21. Lines 425-426 "Even for the same dust microphysical property, the $DRFE_{dust}$ varies significantly according to whether the dust particles are considered spherical or non-spherical in different methods": Particle shape or morphology is microphysical property.

**Reply:** Thank you for your reminding. It has been rewritten as "Even for the same size distributions and the complex refractive index of dust aerosol, the $DRFE_{dust}$ varies significantly according to whether the dust particles are considered spherical or non-spherical in different methods". Please see Lines 476-478 in Page 26 of the revised manuscript.

22. Table 1: Please double-check the data in this table. The ranges "-32.2~-44.3" and "-41.5~-47.4" were obtained due to the SBDART radiative transfer calculation methodology employing different Mie or T-matrix models. The differences cannot be considered as the ranges of $DRFE_{dust}$ variation.

**Reply:** Dust storms occurred on March 9, 11 and 14, 2019 in Sahara Desert, and dust storms occurred on April 9, 23 and 25, 2019 in Taklimakan Desert are chosen as case studies to estimated the $DRFE_{dust}$. Therefore, the $DRFE_{dust}$ variation not only derived from calculation methodology, but also from the dust microphysical property difference in dust storms.

23. Lines 462-463 "The compared results show that the $DRFE_{dust}$ derived from the satellite-based equi-albedo method is closer to that in previous studies with lower uncertainty": I cannot find the evidences that the proposed satellite-based equi-albedo method with lower uncertainty.

**Reply:** The $DRFE_{dust}$ estimated by the satellite-based equi-albedo method is obtained without the dust microphysical properties being assumed, it should have lower uncertainty. The uncertainties of $DRFE_{dust}$ estimated by the satellite-based equi-albedo method are mostly caused by observation errors, and it is hard to estimate the $DRFE_{dust}$ results derived from model simulations. To avoid the ambiguity, we rewrite the sentence as "The compared results show that the $DRFE_{dust}$ derived from the satellite-based equi-albedo method is closer to that in previous studies". Please see Lines 510-511 in Page 27 of the revised manuscript.

24. Line 466 "Therefore, the uncertainties can be evaluated more reasonably.": Please explain and certify "more reasonably".

**Reply:** The SBDART simulation results shows the dust microphysical properties can significantly influence the $DRFE_{dust}$, and the $DRFE_{dust}$ varies significantly according to whether the dust particles are considered spherical or non-spherical in different methods. It can cause significant errors in evaluating the modulating effects of the mineral dust aerosols on climate, and the uncertainty is hard to estimated

(Li et al., 2020;Huang et al., 2009). Instead that, the DRFE$_{dust}$ estimated by the satellite-based equi-albedo method is obtained without the dust microphysical properties being assumed. The uncertainties are mostly caused by observation errors.

The uncertainties of the DRFE$_{dust}$ derived from the equi-albedo method include the instantaneous SW flux error from CERES measurements, the estimation uncertainties of the F$_{clr}$ over the dust storm region, and the uncertainty in the deep blue AOD product. And we consider the uncertainties of the DRFE$_{dust}$ estimated in this paper are reasonable.

**Some technical comments:**

The authors need to read through the manuscript carefully and correct the grammatical errors. I just picked a few of them:

**Reply:** Nanjing Hurricane Translation have review the English language quality of the manuscript.

1.    Lines 58-60: The large spatial variability of aerosols and the lack of an adequate database on their properties makes DRE$_{dust}$ and DRFE$_{dust}$ much difficult to estimated (Satheesh and Srinivasan, 2006).

**Reply:** It has been rewritten as "The large spatial variability of aerosols and the lack of an adequate database on their properties make DRE$_{dust}$ and DRFE$_{dust}$ very difficult to be estimated (Satheesh and Srinivasan, 2006). ". Please see Lines 58-60 in Page 2-3 of the revised manuscript.

2.    Lines 78-80: Thus, the assessment of the SW DRFEdust and microphysical properties of the dust over these regions is important for evaluating regional and global climate changes.

**Reply:** It has been rewritten as "Thus, the assessment on the SW DRFE$_{dust}$ and dust microphysical properties over the Sahara Desert and the Taklimakan Desert is meaningful to evaluate regional and global climate changes. ". Please see Lines 88-90 in Page 3-4 of the revised manuscript.

3.    Lines 156-157: Santa Barbara Disort Atmospheric Radiative Transfer (SBDART) is an RTM that calculates the plane-parallel radiative transfer of the earth-atmosphere system (Ricchiazzi et al., 1998).

**Reply:** It has been rewritten as "Santa Barbara DISORT Atmospheric Radiative Transfer (SBDART) is an RTM that calculates the plane-parallel radiative transfer of the earth-atmosphere system (Ricchiazzi et al., 1998). ". Please see Lines 177-178 in Page 7 of the revised manuscript.

4.    Lines 272-273: To avoid the influence of the LSA and SZA in estimating the DRFEdust, pixels with LSA of 0.16–0.20 and SZA of 32–38 degrees are chosen to derive the DRFE$_{dust}$.

**Reply:** It has been rewritten as "To avoid the influence of the LSA and SZA in estimating the DRFE$_{dust}$, we estimate DRFE$_{dust}$ using pixels with similar LSAs and SZAs. Furthermore, the values of AOD and cloud could also influence the regions we selected. The deep blue algorithm retrieved AOD has large uncertainties in the small value areas. The cloud-free pixels with AOD great than 0.1, and with the LSA of 0.16–0.20 and the SZA of 32–38 degrees are chosen to estimate the DRFE$_{dust}$.". Please see Lines 317-322 in Page 15 of the revised manuscript.

5.    Line 324: The sensitivity test of SW radiative flux at the TOA to changes height of dust layer.

**Reply:** It has been rewritten as "The sensitivity test of SW radiative flux at the TOA to various heights of dust layer. ". Please see Line 377 in Page 20 of the revised manuscript.

6.    Lines 325-326: As Fig.11 shown, the SW radiative flux at the TOA was decreased with the height of dust layer was increased from 0km to 18km.

**Reply:** It has been rewritten as "As shown in Fig. 10, the SW radiative flux at the TOA decreases with the increase of the height of dust layer. ". Please see Lines 378-379 in Page 20 of the revised manuscript.

7.    Lines 326-327: However, the contents of the SW radiative flux change little with the height of dust layer increased (within 1.5Wm$^{-2}$, 0.47%), which is little than CERES observation errors.

**Reply:** It has been rewritten as "However, the contents of the SW radiative flux change little with the increase of the height of dust layer (within 1.5Wm$^{-2}$, 0.47%).". Please see Lines 379-380 in Page 20 of the revised manuscript.

Some figures in the manuscript were hard to read:

8.    Figure 2: The longitude and latitude are not shown in the figure. The regions of the two red boxes do not seem to correspond exactly to the images of MODIS Aqua.

**Reply:** Thanks for the suggestion. We have added the longitude and latitude axis and re-drawn the red boxes to correspond to the images of MODIS Aqua in the figure.

[Figure]

**Figure 12: The research regions and dust storms viewed by MODIS AQUA on 11 March and 9 April 2019.**

Please see Fig.2 in Page 9 of the revised manuscript.

9.    Figures 3,5-10,12-15: Axis labels and legends are very small in these figures.

**Reply:** Thanks for the suggestion. We have re-drawn these figures and increased the character size of axis labels and legends in these figures.

[Figure]

**Figure 13: SW LSA and SZA over Sahara desert and Taklimakan desert derived from AQUA/MODIS.**

[Figure]

**Figure 14: True color images and cloud detections from AQUA/MODIS observations.**

[Figure]

**Figure 15: TOA SW radiative flux derived from AQUA/CERES on March 2019 over Sahara desert and on April 2019 over Taklimakan desert.**

[Figure]

**Figure 16: AOD and DREdust of dust storms on March 2019 over Sahara desert and on April 2019 over Taklimakan desert.**

[Figure]

**Figure 17: Integrated water vapor (g/cm$^2$) from European Centre for Medium-range Weather Forecasts (ECMWF) reanalysis dataset over the Sahara Desert in March 2019 and over the Taklimakan Desert in April 2019.**

[Figure]

**Figure 18:** SBDART simulated clear-sky TOA radiative flux by using integrated water vapor (g/cm$^2$) from ECMWF reanalysis dataset over the Sahara Desert in March 2019 and over the Taklimakan Desert in April 2019.

Please see Fig.3-6 in Page 10-14, Fig.8 in Page 18 and Fig.9 in Page 19 of the revised manuscript.

10. Figure 8: The units of DRFE$_{dust}$ should be added.

**Reply:** Thanks for the suggestion. We have re-drawn the image and added the units of DRFE$_{dust}$.

[Figure]

**Figure 19: DREdust in (a) March 2019 over Tamanrasset and (b) April 2019 over Kashi.**

Please see Fig.7 in Page 16 of the revised manuscript.

11. Figures 14-16: It is recommended to use "Mie" (starting with capital letter) instead of "mie" in these figures.

**Reply:** Thanks for the suggestion. We have re-drawn these Figures and using "Mie" instead of "mie" in these figures.

[Figure]

**Figure 20: Single scattering albedo from (a) the Sahara Desert and (b) the Taklimakan Desert.**

[Figure]

**Figure 21: Asymmetry factor in (a) the Sahara Desert and (b) the Taklimakan Desert.**

[Figure]

[Figure]

**Figure 22: DRFE_dust simulated by the SBDART in (a) Tamanrasset and (b) Kashi.**

Please see Fig.13-15 in Page 23-25 of the revised manuscript.

12. Figures 14,16: The lines in these figures are hard to distinguish.

**Reply:** Thanks for the suggestion. We have re-drawn the Figures, the symbols represents the results derived from different method, and the colors of Line-symbols represents the results in different dust storms. Please see Fig.13 in Page 23 and Fig.15 in Page 25 of the revised manuscript.

The full names of acronyms should be given as they appear for the first time and keep case consistent in the full text:

13. Line 114: the full name of sensor CERES should be given as they appear for the first time in the full text.

**Reply:** Thanks for the suggestion. The full name of sensor CERES have been given in the revised manuscript. Please see Line 124 in Page 5 of the revised manuscript.

14. Figure 3: the full name of "WSA" should be given.

**Reply:** Thanks for the suggestion. We have re-drawn the Figure 3 and provide full name of legends.

[Figure]

**Figure 23: SW LSA and SZA over Sahara desert and Taklimakan desert derived from AQUA/MODIS.**

Please see Fig.3 in Page 10 of the revised manuscript.

15. Lines 194-195 and some other places in the text: "Aqua" and "AQUA" are suggested keeping consistent.

**Reply:** Thank you for the suggestion, we have rewritten "Aqua" as "AQUA" to keep consistent in the revised manuscript. Please see Line 209 in Page 9 of the revised manuscript.

**References**

Christopher, S. A., Chou, J., Zhang, J., Li, X., Berendes, T. A., and Welch, R. M.: Shortwave direct radiative forcing of biomass burning aerosols estimated using VIRS and CERES data, Geophysical Research Letters, 27, 2197-2200, 10.1029/1999gl010923, 2000.

Christopher, S. A., and Zhang, J.: Shortwave Aerosol Radiative Forcing from MODIS and CERES observations over the oceans, Geophysical Research Letters, 29, 6-1-6-4, 10.1029/2002gl014803, 2002.

Christopher, S. A., Zhang, J., Kaufman, Y. J., and Remer, L. A.: Satellite-based assessment of top of atmosphere anthropogenic aerosol radiative forcing over cloud-free oceans, Geophysical Research Letters, 33, https://doi.org/10.1029/2005GL025535, 2006.

Dubovik, O., and King, M. D.: A flexible inversion algorithm for retrieval of aerosol optical properties from Sun and sky radiance measurements, Journal of Geophysical Research: Atmospheres, 105, 20673-20696, 10.1029/2000jd900282, 2000.

Dubovik, O., Sinyuk, A., Lapyonok, T., Holben, B. N., Mishchenko, M., Yang, P., Eck, T. F., Volten, H., Muñoz, O., Veihelmann, B., van der Zande, W. J., Leon, J.-F., Sorokin, M., and Slutsker, I.: Application of spheroid models to account for aerosol particle nonsphericity in remote sensing of desert dust, Journal of Geophysical Research: Atmospheres, 111, 10.1029/2005jd006619, 2006.

García, O. E., Díaz, J. P., Expósito, F. J., Díaz, A. M., Dubovik, O., Derimian, Y., Dubuisson, P., and Roger, J. C.: Shortwave radiative forcing and efficiency of key aerosol types using AERONET data, Atmos. Chem. Phys., 12, 5129-5145, 10.5194/acp-12-5129-2012, 2012.

Guirado, C., Cuevas, E., Cachorro, V. E., Toledano, C., and Frutos, A.: Aerosol characterization at the Saharan AERONET site Tamanrasset, Atmospheric Chemistry and Physics, 14, 2014.

Heald, C. L., Ridley, D. A., Kroll, J. H., Barrett, S. R. H., Cady-Pereira, K. E., Alvarado, M. J., and Holmes, C. D.: Contrasting the direct radiative effect and direct radiative forcing of aerosols, Atmos. Chem. Phys., 14, 5513-5527, 10.5194/acp-14-5513-2014, 2014.

Hsu, N. C., Tsay, S. C., King, M. D., and Herman, J. R.: Aerosol properties over bright-reflecting source regions, IEEE Transactions on Geoscience & Remote Sensing, 42, 557-569, 2004.

Huang, J., Fu, Q., Su, J., and Tang, Q.: Taklimakan dust aerosol radiative heating derived from CALIPSO observations using the Fu-Liou radiation model with CERES constraints, Atmospheric Chemistry and Physics Discussions, 2009.

Jose, S., Gharai, B., Rao, P. V. N., and Dutt, C. B. S.: Satellite-based shortwave aerosol radiative forcing of dust storm over the Arabian Sea, Atmospheric Science Letters, 17, 43-50, https://doi.org/10.1002/asl.597, 2016a.

Jose, S., Gharai, B., Rao, P. V. N., and Dutt, C. B. S.: Satellite‐based shortwave aerosol radiative forcing of dust storm over the Arabian Sea, Atmospheric Science Letters, 17, 43-50, 2016b.

Lewis, P., and Barnsley, M.: Influence of the sky radiance distribution on various formulations of the Earth surface albedo, Proc. Conf. Phys. Meas. Sign. Remote Sens., 1994.

Li, F., Vogelmann, A. M., and Ramanathan, V.: Saharan Dust Aerosol Radiative Forcing Measured from Space, Journal of Climate, 17, 2558-2571, 10.1175/1520-0442(2004)017<2558:SDARFM>2.0.CO;2, 2004.

Li, L., Li, Z., Chang, W., Ou, Y., Goloub, P., Li, C., Li, K., Hu, Q., Wang, J., and Wendisch, M.: Aerosol solar radiative forcing near the Taklimakan Desert based on radiative transfer and regional meteorological simulations during the Dust Aerosol Observation-Kashi campaign, Atmos. Chem. Phys.,

20, 10845-10864, 10.5194/acp-20-10845-2020, 2020.

Lin, C., Guang-Yu, S., Ling-Zhi, Z., and Sai-Chun, T.: Assessment of Dust Aerosol Optical Depth and Shortwave Radiative Forcing over the Northwest Pacific Ocean in Spring Based on Satellite Observations, Atmospheric and Oceanic Science Letters, 2009.

Ramaswamy, V.: IPCC Third Assessment Report - Chapter 6 - Radiative Forcing of Climate Change, 2006.

Remer, L. A., Kaufman, Y. J., Tanré, D., Mattoo, S., Chu, D. A., Martins, J. V., Li, R. R., Ichoku, C., Levy, R. C., Kleidman, R. G., Eck, T. F., Vermote, E., and Holben, B. N.: The MODIS Aerosol Algorithm, Products, and Validation, Journal of the Atmospheric Sciences, 62, 947-973, 10.1175/JAS3385.1, 2005.

Satheesh, S. K., and Ramanathan, V.: Large differences in tropical aerosol forcing at the top of the atmosphere and Earth's surface, Nature, 405, 60-63, 10.1038/35011039, 2000.

Schaaf, C. B., Gao, F., Strahler, A. H., Lucht, W., Li, X., Tsang, T., Strugnell, N. C., Zhang, X., Jin, Y., Muller, J.-P., Lewis, P., Barnsley, M., Hobson, P., Disney, M., Roberts, G., Dunderdale, M., Doll, C., d'Entremont, R. P., Hu, B., Liang, S., Privette, J. L., and Roy, D.: First operational BRDF, albedo nadir reflectance products from MODIS, Remote Sensing of Environment, 83, 135, 10.1016/s0034-4257(02)00091-3, 2002.

Tian, L., Zhang, P., and Chen, L.: Estimation of the Dust Aerosol Shortwave Direct Forcing Over Land Based on an Equi-albedo Method From Satellite Measurements, Journal of Geophysical Research: Atmospheres, 124, 8793-8807, 10.1029/2019JD030974, 2019.

Wielicki, B. A., Barkstrom, B. R., Baum, B. A., Charlock, T. P., Green, R. N., Kratz, D. P., Lee, R. B., Minnis, P., Smith, G. L., Takmeng, W., Young, D. F., Cess, R. D., Coakley, J. A., Crommelynck, D. A. H., Donner, L., Kandel, R., King, M. D., Miller, A. J., Ramanathan, V., Randall, D. A., Stowe, L. L., and Welch, R. M.: Clouds and the Earth's Radiant Energy System (CERES): algorithm overview, IEEE Transactions on Geoscience and Remote Sensing, 36, 1127-1141, 1998.

Xia, X., and Zong, X.: Shortwave versus longwave direct radiative forcing by Taklimakan dust aerosols, Geophysical Research Letters, 36, 10.1029/2009gl037237, 2009.

Zhang, J., and Christopher, S. A.: Longwave radiative forcing of Saharan dust aerosols estimated from MODIS, MISR, and CERES observations on Terra, Geophysical Research Letters, 30, https://doi.org/10.1029/2003GL018479, 2003.

---

## Author Comment (AC4)

**Responses to the comments**

Dear Reviewer,

Thank you for carefully reviewing our manuscript. The comments have helped us to improve the paper. We have classified these comments into 4 groups. The comments group 1 (includes comments 2, 3, 5, 12, 15, 16 and 39) is related with the selected study area and the matched AERONET sites in figure 3, 4, 5, 6, 8, 9 of the manuscript. The comments group 2 (comments 6 and 10) is related with the detailed information of the aerosol optical properties used in Radiative Transfer Mode (RTM) for dust aerosol Direct Radiative Effect ($DRE_{dust}$) estimation. The comments group 3 is related with the AERONET data using conditional. And comments group 4 is related to clear and coherent expression in the manuscript.

Following these comments and suggestions, we take a lot of efforts to optimize the structure of the manuscript, add the explanation on how to calculation the optical characteristics of the dust aerosol, analysis the cloud effect on the two AERONET sites, re-draw the most of the imaging in the article to meet the reviewer's suggestion, and modify the inaccurate information to avoid the ambiguity.

We response every comments sentence by sentence. Please find the comments in blue italics and our reply in black.

1.    This paper determines the TOA direct radiative effect (DRE) and the so-called direct radiative forcing efficiency (DRFE) of dust for two dust storms by using CERES SSF data. One of the dust storms occurs near an AERONET site in Tamanrasset (Africa) and the other occurs near Kashi, India. The authors note that the mineralogy of dust is different for these two regions. The authors would like to study the impact of mineralogy on the dust radiative effect, so they constrain their analysis to a single land surface albedo (LSA, from MODIS) and a single solar zenith angle (SZA). Since the authors are looking at multiple locations with the same surface albedos, they call this the equi-albedo method. The authors also focus on DRFE instead of DRE in order to eliminate the effect of column loading. Since the authors have constrained most of the parameters that affect TOA DRFE, they attribute DRFE differences between the two storms to differences in dust minerology. Thus, the authors investigate further by analyzing the differences in microphyical dust properties inferred at the AERONET sites during the dust storms. This is an interesting idea, but the analysis is not terribly convincing.

**Reply:** We try our best to response the each suggestion from the reviewer. We optimize the structure of the manuscript, add the explanation on how to calculation the optical characteristics of the dust aerosol, analysis the cloud effect on the two AERONET sites, re-draw the most of the imaging in the article to meet the reviewer's suggestion, and modify the inaccurate information to avoid the ambiguity. We hope these responses could improve this article's quality and could enhance the scientific persuasion.

One information from the reviewer's comments is incorrect and I have to correct. The Kashi site we selected in this paper located in the Northwest part of China, not in India.

2.    The paper is telling a pretty reasonable story until the arrival of Figure 7. Here, the data of the earlier maps is reduced to a few 1x1 deg areas. Undaunted, the authors discuss how    "The high dust aerosol loading regions show significant negative radiative forcing..." (lines 267-268). I guess that the reader is supposed to scroll between Figs 5 & 7 to confirm this, but requiring a reader to scroll between two figures does not generally convince anyone of anything.

**Reply:** I totally agree with the reviewer's suggestion. To make the user compare the results easily, we have re-plotted the $DRE_{dust}$ and AOD in the same image according to the reviewer's suggestion. The readers do not need to scroll between two figures anymore.

In previous study, we found that $DRE_{dust}$ is significantly influenced by LSA and SZA (Tian et al., 2019). To avoid the influence of the LSA and SZA in estimating the $DRFE_{dust}$, we estimate $DRFE_{dust}$ using pixels with similar LSA and SZA.

[Figure]

Figure 1: AOD and DREdust of dust storms over the Sahara Desert in March 2019 and over the Taklimakan Desert in April 2019.

Fig. 1 shows that AOD and DRE$_{dust}$ plotted in the same image, the high dust aerosol loading regions show significant negative radiative forcing. It indicates that the dust aerosol loading is negatively correlated with the DRE$_{dust}$ in these dust storm events. Thus, dust aerosols have a negative radiative effect in the SW spectrum. We also plotted all the available data of DRE$_{dust}$ in the distribution maps, and put black borders around the chosen pixels. Please see Fig. 6 and Lines 301-324 in Page 14 -15 of the revised manuscript.

Moreover, reader could also confirm the "The dust aerosol loading regions show significant negative radiative forcing" from the Table 1 and Fig. 7 in the revised manuscript.

3.     The frustrating part is that there is no need for the maps in Fig 7 to be so sparse -- the data is available. Now, I realize that the authors want to focus on 1x1 regions that are constrained by LSA and SZA, but there are other ways of dealing with this. For instance, one can include data for the entire maps in Fig 7, and then put a black border around the few 1x1 grids of interest. This will allow the authors to discuss the whole map (which they often do for these sparse maps) as well as the regions of

interest. Furthermore, include the same borders in Figs 3,4,5,6. This will help the reader to understand the cloud fields, AOD, and TOA SW radiative flux in the 1x1 regions of interest. Since these regions of interest are only constrained by surface albedo and SZA, they can be easily introduced in Fig 3. If you do it this way, this will allow the reader to see the patterns of DRE in Fig 7 and make the paper a whole lot more interesting. Same thing with Figs 9&10 -- show the IWV and SBDART fluxes everywhere, but outline the 1x1 regions of interest with borders. Finally, include the Tamaarasset and Kashi sites in all of the maps (including figures 4, 6, 7, 9, and 10). The authors are trying to link surface measurements at these two sites to the dust storms observed in the satellite data, and the link is very weak because they do not show the location of these sites on many of these maps.

**Reply:** This suggestion is pretty valuable. We have re-plotted all the available data of DRE$_{dust}$ (please see the Fig. 1 in the reply of comments 2), integrated water vapor and SBDART derived SW fluxes in these distribution maps, and put black borders around the chosen pixels in these figures.

[Figure]

Figure 2: Integrated water vapor (g/cm$^2$) from European Centre for Medium-range Weather Forecasts (ECMWF) reanalysis dataset over the Sahara Desert in March 2019 and over the Taklimakan Desert in April 2019.

Fig. 2 shows the integrated water vapor from ECMWF reanalysis dataset over the Sahara Desert

in March 2019 and over the Taklimakan Desert on April 2019. The grids surrounded by black border are the chosen pixels to estimate the DRFE$_{dust}$. The integrated water vapor varies little over different research areas, and the mean differences of chosen pixels are 0.51g/cm$^2$ and 0.18g/cm$^2$ over the Sahara Desert and the Taklimakan Desert, respectively.

[Figure]

Figure 3: SBDART simulated clear-sky TOA radiative flux by using integrated water vapor (g/cm$^2$) from ECMWF reanalysis dataset over the Sahara Desert in March 2019 and over the Taklimakan Desert in April 2019.

Fig. 3 shows the SBDART simulated clear-sky TOA radiative flux by using the integrated water vapor from ECMWF reanalysis dataset over the Sahara Desert in March 2019 and over the Taklimakan Desert in April 2019, and the grids surrounded by black border are the chosen pixels to derive the DRFE$_{dust}$. The regional mean differences of TOA radiative flux are 2.21% and 0.85% over the Sahara Desert and the Taklimakan Desert, respectively.

Moreover, we also marked the location of Tamanrasset and Kashi sites in all of the maps. Please see Fig. 3 in Page 10, Fig. 4 in Page 11, Fig. 5 in Page 13, Fig. 6 in Page 14, Fig. 8 in Page 18 and Fig. 9 in Page 19 of the revised manuscript.

4.    Is is also very odd that the 1x1 regions with data vary from day to day. Perhaps more odd, sometimes some of the boxes vary, but others do not. Since these regions are selected based upon LSA and SZA constraints, why don't the same regions show up on Mar 9, 11, and 14 at Tamanrasset and on Apr 9, 23, and 25 at Kashi? I haven't kept up on the MODIS albedo products, but it used to be produced every 2 weeks. Thus, if LSA is the same, the only parameter that will move these boxes around is the SZA. An explanation about why the SZA apparently varies so much on the different days would be helpful.

**Reply:** To avoid the influence of the LSA and SZA in estimating the $DRFE_{dust}$, we estimate $DRFE_{dust}$ using pixels with similar LSA and SZA. Furthermore, the values of AOD and cloud also influenced the regions we selected these chosen pixels. The deep blue algorithm retrieved AOD have large uncertaintites in the small value area (Sayer et al., 2014). Thus, the chosen pixels should have AOD great than 0.1, and in clear-sky condition.

The LSA would not have great changes in 2 weeks, and the MODIS Collection6 albedo product dataset (MCD43C3) (Schaaf et al., 2011;Schaaf et al., 2002;Schaaf et al., 2008) provide broad band land surface albedo every 2 weeks. The movements of these chosen pixels were mainly caused by the variety of SZA, the value of AOD and cloud coverage.

[Figure]

Figure 4: SW LSA and SZA over the Sahara Desert and the Taklimakan Desert derived from AQUA/MODIS.

Fig. 4 shows the LSA and the SZA observed by the AQUA satellite over Sahara Desert (Fig. 4(a1)-(a3) and Fig. 4(b1)-(b3)) and Taklimakan Desert (Fig. 4(c1)-(c3) and Fig. 4(d1)-(d3)) during dust storms. The distribution maps shown SZA differences in these dust storms. It is because the time of satellite scanning the same place are varies in each days. And the AOD values (please see Fig. 6 in the revised manuscript) and cloud coverage (please see Fig. 4 in the revised manuscript) also changes in these dust storms.

Therefore, the selected pixels to derive the DRFE$_{dust}$ varies obviously in these dust storms. We have the pretty strict threshold to sample the data for DRFE$_{dust}$ estimation.

5. The other component of this paper is computing TOA DRE and DRFE from the microphysical properties at two AERONET sites. The purpose is to link the two techniques together (satellite and surface retrieval computaions), but the link is weak since the reader does not even know if the AERONET sites a located within any of the regions with data in Figs 7,9, and 10, or how DRFE varies across the maps. It would have been interesting to see a map of DRFE for the entire maps.

**Reply:** We have marked the location of Tamaarasset and Kashi sites in Fig 3, Fig 4, Fig 5, Fig 6, Fig 8 and Fig 9 in the revised manuscript.

Moreover, we also would like to explain that, the aerosols measured in Tamanrasset can represent the pure dust aerosols from the Sahara Desert (Guirado-Fuentes et al., 2014), and Kashi represents a place affected by dust aerosols transported from the Taklimakan Desert (Li et al., 2020), the dust aerosols observed in Tamanrasset and Kashi sites are typical samples of the dust aerosols from these two deserts. We supposed the microphysical properties of dust aerosols in Sahara desert and Taklimakan desert are similar with dust aerosols in Tamanrasset and Kashi, and SW DRFE$_{dust}$ are mainly determined by microphysical properties of dust aerosols, LSA and SZA. To avoid the influence of the LSA and SZA in estimating the DRFE$_{dust}$, we estimate DRFE$_{dust}$ using pixels with similar LSA and SZA. Therefore, although AERONET sites not always located within the selected pixels, it would have few influences on the estimation of DRFE$_{dust}$.

6. Putting collocation aside, the details of the microphysical calculations are missing. The authors use SBDART for broadband computations, but they only have optical properties at four wavelengths. How do they extrapolate the AERONET refractive indices throughout the SW? The methodology is sprinkled throughout the paper, and is sometimes inconsistent. For instance, the authors state that SSA and ASY are calculated using spherical and non-spherical methods -- how? Do the authors do this, or does SBDART take care of this? On lines 240-242 the authors say that AERONET computes SSA and ASY. Later (on line 153) they say that the "NASA-GISS code is used to calculate the optical properties of the spherical particles and the ellipsoidal particles." On line 375 we're back to AERONET.

**Reply:** In this study, we accessed aerosol microphysical properties data (include volume size distribution and the refractive index of the dust aerosol), and aerosol optical properties were calculated based on aerosol microphysical properties and T-matrix method. The T-matrix codes are accessed from the National Aeronautics and Space Administration (NASA) Goddard Institute for Space Studies (GISS) group (https://www.giss.nasa.gov/staff/mmishchenko/t_matrix.html). The codes are directly applicable to spheroids and finite circular cylinders, and spheroids are formed by rotating an ellipse about its minor (oblate spheroid) or major (prolate spheroid) axis (Mishchenko and Travis, 1998):

$$r(\theta, \phi) = a\left[sin^2\theta + \frac{a^2}{b^2}cos^2\theta\right]^{-1/2}$$

where $\theta$ is the polar angle, $\phi$ is the azimuth angle, b is the rotational (vertical) semi-axis, and a is the horizontal semi-axis. The shape and size of a spheroid can be conveniently specified by the aspect ratio (a/b). The aspect ratio is greater than 1 for oblate spheroids, smaller than 1 for prolate spheroids, and equal to 1 for spheres. Therefore, Mie scattering method can be regarded as a special case of the T-matrix method. The dust particles are assumed to sphere (aspect ratio equal to 1) and ellipsoid (aspect ratio equal to 0.8) for dust aerosol optical properties calculating. The results shows the dust aerosol optical properties were difference in particles shape assuming, and ellipsoid (aspect ratio equal to 0.8) results is closer to those estimated by AERONET inversion aerosol optical products and the satellite observations, that is indicates most dust aerosols are non-spherical in the natural environment. We have added the description of spherical and non-spherical methods in the revised manuscript. Please see Lines 164-175 in Page 7 of the revised manuscript.

In SBDART model, user defined aerosol spectral dependence by few wavelengths points, the aerosol optical properties is extrapolated to other wavelengths using a power law (Ricchiazzi et al., 1998). Therefore, aerosol properties measured at four wavelengths are extrapolated so that flux calculations can be made in any desired wavelength across the shortwave spectrum (McComiskey et al., 2021). We have added the detailed description for the aerosol properties spectral extrapolate method.

Please see Lines 191-196 in Page 8 of the revised manuscript.

7. The authors do not discuss details of their datasets. For instance, do they use AERONET Version 2 or Version 3? Level 1.5 or Level 2? Version 2 is no longer available, but the authors may be using previously-downloaded data. Also, the Version 3 retrievals at Tamanrasset never made it to Level 2, indicating that the data did not make it through the new cloud screening process. This is important because cloud contamination could easily confound their conclusions about this part of the paper.

**Reply:** In this study we use AERONET Version 3 Level 1.5 data. We have made efforts to prove these data were not influenced by cloud.

In this paper, the $DRFE_{dust}$ of the Taklimakan Desert is estimated with the same dust properties referring to the works of Li et al. (Li, L., Li, Z., Chang, W., Ou, Y., Goloub, P., Li, C., Li, K., Hu, Q., Wang, J., and Wendisch, M.: Aerosol solar radiative forcing near the Taklimakan Desert based on radiative transfer and regional meteorological simulations during the Dust Aerosol Observation-Kashi campaign, Atmos. Chem. Phys., 20, 10845-10864, 10.5194/acp-20-10845-2020, 2020) (Li et al., 2020). In the paper, Li et al gives the sky conditions using full-sky visible images on 9 and 24 April, 2019 at Kashi (Fig. 5).

[Figure]

[Figure]

Figure 5: Full-sky visible images on 9 and 24 April, 2019 at Kashi (Li et al., 2020).

Fig. 5 gives the sky conditions on 9 and 24 April, 2019 at Kashi, these images clearly shows Kashi have dust events and not covered by clouds on 9 and 24 April, 2019.

Although we did not get the full-sky visible images at Tamanrasset, we can also access the sky conditions of Tamanrasset from satellite observations.

[Figure]

Figure 6: True color images and cloud detections from AQUA/MODIS observations.

Fig. 6 gives the true color images (Fig. 6(a1)-(a3) over Sahara Desert, and Fig. 6(c1)-(c3) over Taklimakan Desert) and cloud detections (Fig. 6(b1)-(b3) over Sahara Desert, and Fig. 6(d1)-(d3) over Taklimakan Desert) from AQUA/MODIS cloud mask products (MYD035). From the true color images and cloud detections we can see that, Tamanrasset and Kashi were not covered by clouds during these dust storm events.

Therefore, we can guarantee that these data were not polluted by cloud. We have explained in the in the manuscript. Please see Fig 4 and Lines 232-240 in Page 10-11 the revised manuscript.

8.    The authors seem to have some misconceptions about SSA. On line 382, they state: "A high SSA is correlated with low real parts of the complex refractive index, while a strong absorption is correlated with a high imaginary part of the complex refractive index. Together with the size distribution, real parts of the complex refractive index can determine the magnitude of the SSA."    <--- This is incorrect; the SSA is largely determined by the *imaginary* refractive index and the size distribution.

**Reply:** Thank you for helping us to check out the mistake. The correct meaning is "The size distribution and the complex refractive index can codetermine the magnitude of the SSA.". We have corrected this mistake in the manuscript. Please see Line 436-437 in Page 23 of the revised manuscript.

9.  Story does not flow and jumps around. The writing is pretty sloppy, as evidenced by the long list of issues below.

**Reply:** Thank you for the criticisms, we have taken a lot of efforts to make the writing more concise and clear.

10.  Line 154: Why is the particle aspect ratio set to 0.8? Is this based upon a literature value? Need to tell the reader.

**Reply:** In this study, the RTM results were used to theoretical verification of $DRFE_{dust}$ derived from satellite observation. The dust particles are assumed to sphere (aspect ratio equal to 1) and ellipsoid (aspect ratio equal to 0.8) for dust aerosol optical properties calculating. The results shows the dust aerosol optical properties were difference in particle shape assuming, and ellipsoid (aspect ratio equal to 0.8) results is closer to those estimated by AERONET inversion aerosol optical products and the satellite observations, that is indicates most dust aerosols are non-spherical in the natural environment. There may have better assumptions of aerosol particle shapes to calculate the aerosol optical properties closer to the real values, and that need further works on observation and research. In this study, we use sphere (aspect ratio set as 1.0) and ellipsoid (aspect ratio set as 0.8) to discuss the aerosol optical properties were difference in particles shape assuming.

We have added the detailed information about the particle aspect ratio setting in the manuscript. Please see Lines 170-175 in Page 7 of the revised manuscript.

11.  Line 193: There is no Version 3 Level 2 data in March at the Tamanrasset site. So it is possible (probable?) that this data is contaminated by clouds. This could contribute significantly to the retrieval differences between the two sites. An explanation is necessary.

**Reply:** Thank you for the suggestion, the suggestion also made in comments 7, the full-sky visible images from Li et al. (2020) (Li et al., 2020), true color images and cloud detections from AQUA/MODIS observations clearly shows Tamanrasset and Kashi were not covered by clouds during these dust stroms. Therefore, we can guarantee that these data were not influenced by cloud.

We have added the explanation in the in the manuscript. Please see Fig. 4 and Lines 232-240 in Page 10-11 the revised manuscript.

12.  Line 254: "The TOA SW radiative flux distribution shows the highest value over cloud conditions" ...How can I tell this from Fig 6?... Tell the reader that they can find the clouds in Fig 3. Better yet, design the paper so that you can combine these flux figures with the Fig 3 images. So, one figure would contain the left panels of Figs 3 & 6 and another figure would contain the right panels of Figs 3 & 6. That way the cloud images are side by side with the flux figures.

**Reply:** Thank you for the suggestion, we have added the cloud detection results in Fig. 4, and we also told the reader the cloud distribution maps can be found from Fig. 4 in the revised manuscript. Please see Fig. 4 in Page 11 and Lines 291-292 in Page 13 of the revised manuscript.

13.  Line 256: Sentence unclear.

**Reply:** We have revised this sentence in the manuscript. It has been rewritten as "Following the equi-albedo method (Tian et al., 2019), the $F_{clr}$ and $DRE_{dust}$ over the Sahara Desert and the Taklimakan Desert can be estimated based on the measurements from MODIS and CERES both aboard on the AQUA satellite.". Please see Lines 298-300 in Page 14 of the revised manuscript.

14.  Line 257: "Thus, dust aerosols have a negative radiative effect in the SW spectrum." ...Here again -- how do I get this from Fig 6, where all numbers are positive? If you want to discuss radiative effect, why not show radiative effect in the figure? ...I see you have rad effect in Fig 7. Why not delay this discussion until then?

**Reply:** Dust aerosols have higher SW albedo than land surface albedo in clear-sky conditions, and dust aerosols reflect more SW radiation to TOA. $DRE_{dust}$ was defined as the radiative fluxes difference between clear ($F_{clr}$) and dust loading ($F_{dust}$) conditions (Garrett and Zhao, 2006;Christopher et al., 2000;Ramanathan et al., 1989).

$$DRE_{dust} = F_{clr} - F_{dust}$$

Therefore dust aerosols have a negative radiative effect in the SW spectrum. It also can be founded in Fig. 6 in the revised manuscript, we delayed this discussion behind Fig. 6 in the revised manuscript. Please see Lines 312-313 in Page 15 of the revised manuscript.

15.  Line 267: "The high dust aerosol loading regions show significant negative radiative forcing" ...a little difficult to conclude this with so sparse data in Fig 7.

**Reply:** Thank you for the suggestion, the suggestion also made in comment 3. We have re-plotted the $DRE_{dust}$ and AOD distribution maps in the same image, and all the available data of $DRE_{dust}$ were given in the new figure. The selected pixels to derive the $DRFE_{dust}$ were surrounded by black borders in these figures. Please see Fig.6 and Lines 304-313 in Page 14-15 of the revised manuscript.

16.  Line 308: "The integrated water vapor varies little over research areas,..." -- here again, how can the reader know when you only show a few points in Fig 9? Surely ECMWF provides more H2Ov than this?

**Reply:** Thank you for the suggestion, the suggestion also made in comment 3. We have re-plotted all the available data of integrated water vapor and SBDART derived SW fluxes in these distribution maps, and put black borders around the chosen pixels in these figures. Please see Fig. 8 in Page 18 and Fig. 9 in Page 19 of the revised manuscript.

17.  Lines 382-384: Incorrect.

**Reply:** Thank you for helping us to check out the mistake, we have revised in the manuscript. It has been rewritten as "The size distribution and the complex refractive index can codetermine the magnitude of the SSA.". Please see Line 436-437 in Page 23 of the revised manuscript.

18.  Line 415: "As shown in Fig. 16, with higher aerosol scattering (higher SSA) and higher backward scattering coefficients (lower ASY), the negative DRFEdust from Kashi is more significant." <-- I don't see how this follows from Fig 16, as SSA is not even mentioned in the Figure.

**Reply:** We have revised the sentence and added the state of SSA and ASY in the manuscript. It has been rewritten as "As shown in Fig. 15, with higher aerosol scattering (higher SSA in Fig. 13) and higher backward scattering (lower ASY in Fig. 14), the negative $DRFE_{dust}$ from Kashi is more significant.". Please see Lines 466-468 in Page 25-26 the revised manuscript.

19.  Throughout the paper the authors assume that all aerosols are dust, both for the satellite data set and for the surface measurements. That's ok, but it needs to be stated.

**Reply:** We have added the explanation of the assume that all aerosols are dust aerosol in dust storms. It has been written as "Since the Sahara Desert and the Taklimakan Desert are free of industrial activities, the major aerosol over the desert areas is dust aerosol, and the anthropogenic and marine aerosols have little contribution to the total AOD, especially during dust storm episodes. Thus, we directly use the AOD retrieved by MODIS to estimate $DRFE_{dust}$ during dust storms in this study.". Please see Lines 256-259 in Page 12 of the revised manuscript.

20. Line 45: Anderson (2005) does seem to claim ownership of this idea, but forcing efficiency dates back to at least Satheesh (2000) papers. There are many others, but at least Satheesh (2000) needs to be added.

**Reply:** Thank you for the suggestion. We have revised the sentence and added the citations in the manuscript. Please see Lines 44-47 in Page 2 of the revised manuscript.

21. Line 193: This is confusing. The authors provide two dates here, but figs 3&4 provide 3 dates for each site. They have already covered Fig 2 in the previous paragraph -- why are these first two sentences even located in this paragraph?? This should really be a lead-in for the previous paragraph.

**Reply:** We have revised it in the in the manuscript. Please see Lines 222-228 and Fig. 3 in Page 9-10 of the revised manuscript.

22. Line 220: Retried?

**Reply:** Thank you for helping us to check out the mistake. This is a typing mistake, is should be "Retrieved". We have revised it in the manuscript. Please see Line 254 in Page 11 of the revised manuscript.

23. Line 236: need to cite original AERONET paper; Holben et al, 1998.

**Reply:** Thank you for the suggestion, we have added the citation of Holben et al (1998). Please see Line 278 in Page 12 of the revised manuscript.

24. Line 281: "According to the definition, the DRFEdust represents the DREdust of 281 a certain AOD at per unit area..." -- not just any AOD, but at AOD = 1, right? Should state that.

**Reply:** The $DRFE_{dust}$ represents the $DRE_{dust}$ of per unit aerosol optical depth (AOD), which means the efficiency of the dust aerosol that affects the net radiative flux of solar radiation. We have revised this sentence in the manuscript. It has been rewritten as "According to the definition, the $DRFE_{dust}$ represents the $DRE_{dust}$ of per unit AOD during these storms in the dust source regions.". Please see Lines 329-330 in Page 16 of the revised manuscript.

25. Line 285: AOD wavelength should be mentioned in caption.

**Reply:** We have added the describtion of the wavelength of AOD (0.55μm). Please see Line 334 in Page 16 of the revised manuscript.

26. Line 289: Two numbers, one location.

**Reply:** Thank you for helping us to check out the mistake. This is a typing mistake, it is should be "$DRFE_{dust}$ of the dust storms is $-39.6$ $Wm^{-2}\tau^{-1}$ over Tamanrasset and $-48.6$ $Wm^{-2}\tau^{-1}$ over Kashi". We have corrected the mistake. Please see Lines 336-338 in Page 16 of the revised manuscript.

27.  Line 285: Is the regression forced through the origin? If not, state the offset so we know. Early papers often had an offset.

**Reply:** The regression did not forced through the origin, the regression offset are -0.818 Wm$^{-2}$ and -1.602 Wm$^{-2}$ over Sahara Desert and Taklimakan Desert separately. The small offset indicates that method we used in this paper is effective, and the DRFE$_{dust}$ estimated from satellite observation has good reliability.

**Reply:** We have revised this sentence in the manuscript. It has been rewritten as "In order to estimate the uncertainties caused by the variation of integrated water vapor over chosen pixels, we have calculated the SW radiative flux at the TOA under different integrated water vapor based on the SBDART model.". Please see Lines 360-362 in Page 18 of the revised manuscript.

29.  Line 324: This figure would be more intiutive if the independent variable was put on the x-axis.

**Reply:** Thank you for the suggestion. We have re-plotted the figure, and put the height on the x-axis.

[Figure]

Figure 7: The sensitivity test of SW radiative flux at the TOA to various heights of dust layer.
Please see Fig. 10 in Page 20 of the revised manuscript.

30.  Line 327: "...which is little than CERES observation errors." <---------- nonsensical

**Reply:** The radiative fluxes at TOA are derived from the CERES radiance measurements in three broad-band channel, using empirical Angular Distribution Models (ADMs). The ADMs are a function of varying scene types, such as land, ocean, cloud cover, aerosols, etc. Research shows that the uncertainty of TOA instantaneous shortwave flux is about 1.6% (4.5Wm-2) over clear-sky land, and about 2.7% (8.4Wm−2) over land under all-sky conditions (Su et al., 2015), and the overall bias in

monthly regional albedos based on ADMs are < 4% (Loeb et al., 2021). Therefore, we suppose the SW fluxes in differences of 1.5Wm$^{-2}$ (0.47%) is less than CERES observation errors.

We have added the explanation of this sentence in the manuscript. Please see Lines 378-380 in Page 20 of the revised manuscript.

31.  Lines 330-332: again, nonsensical.

**Reply:** We had not exactly understood the meaning of "nonsensical". The uncertainty discussions are important for evaluation of the results. In our previous articles (Tian et al., 2019), reviewers highly recommend us to add the uncertainties discussions. The each source of the uncertainties were important and can be derived from satellite observation errors (Sayer et al., 2014;Su et al., 2015;Loeb et al., 2021) and sensitive tests (Tian et al., 2019). And the estimation of the total uncertainty is following previous research (Zhang et al., 2005).

32.  Line 349: why the name change from tamanrasset and kashi to Sahara and Takliman?

**Reply:** The aerosols measured in Tamanrasset can represent the pure dust aerosols from the Sahara Desert (Guirado-Fuentes et al., 2014), and Kashi represents a place affected by dust aerosols transported from the Taklimakan Desert (Li et al., 2020), the dust aerosols observed in Tamanrasset and Kashi sites are typical samples of the dust aerosols from these two deserts.

We have revised all the confusing sentences in the revised manuscript. Please see Lines 222-223 in Page 9, Lines 230-233 in Page 12, Lines 287-290 in Page 13, Lines 297-305 in Page 14, Lines 353-358 in Page 18 and Lines 364-369 in Page 19 of the revised manuscript.

33.  Line 349: why is Kashi associated with the Sahara desert in panel d?

**Reply:** Thank you for helping us to check out the mistake.

(a) Real parts of the complex refractive index over Sahara Desert

(b) Real parts of the complex refractive index over Taklimakan Desert

(c) Imaginary parts of the complex refractive index over Sahara Desert

[Figure]

(d) Imaginary parts of the complex refractive index over Taklimakan Desert

[Figure]

Figure 8: Real and imaginary parts of the dust complex refractive index from the Sahara Desert and the Taklimakan Desert.

We have corrected the mistake in the manuscript. Please see Fig. 11 in Page 21 of the revised manuscript.

34. Line 359: "The volume size distribution of dust aerosols clearly shows the particle size difference between dusty and clear-sky days." Authors need to point to a figure when making a statement like this. can not conclude this from the most recent figure (fig 12).

**Reply:** It can not be concluded this from Fig. 12, we deleted this sentence in the manuscript. Please see Line 416 in Page 22 of the revised manuscript.

35. Line 359-362: these first 3 sentences make no sense b/c you have not told the reader what you are talking about!

**Reply:** We rewrite these sentence in the manuscript. Please see Lines 416-421 in Page 22 of the revised manuscript.

36. Line 365: units

**Reply:** We have added units (μm) in the manuscript. It has been rewritten as "Most maximum dust aerosol size distribution peaks at the radius of 1.71μm in Tamanrasset and 2.24μm in Kashi.". Please see Lines 417-418 in Page 22 of the revised manuscript.

37. Lines 359-367: What is the point of this paragraph?

**Reply:** Here describes the characteristics of Dust aerosol size distribution over the Sahara Desert and

the Taklimakan Desert. We have deleted redundant statements to make the writing more concise and clear. Please see Lines 416-421 in Page 22 of the revised manuscript.

38. Figure 6: Tell us in the caption that this is CERES. Ideally, include some details about the particular CERES product. Most folks won't read more than the captions in your paper.
**Reply:** We have added the captions of the figure. It has been rewritten as "Figure 5: Figure 5: TOA SW radiative flux derived from AQUA/CERES over the Sahara Desert on March 2019 and over the Taklimakan Desert on April 2019.". Please see Fig. 5 and Lines 287-288 in Page 13 of the revised manuscript.

39. Figure 10: Again -- why not the whole map?
**Reply:** Thank you for the suggestion, the suggestion also made in comment 3 and 16. We have re-plotted all the available data of integrated water vapor and SBDART derived SW fluxes in these distribution maps, and put black borders around the chosen pixels in these figures. Please see Fig. 8 in Page 18 and Fig. 9 in Page 19 of the revised manuscript.

40. Fig 11: independent variable should be on x-axis.
**Reply:** Thank you for the suggestion, the suggestion also made in comment 29. We have re-plotted the figure, and put the height on the x-axis.
    Please see Fig. 10 in Page 20 of the revised manuscript.

**References**

Christopher, S. A., Chou, J., Zhang, J., Li, X., Berendes, T. A., and Welch, R. M.: Shortwave direct radiative forcing of biomass burning aerosols estimated using VIRS and CERES data, Geophysical Research Letters, 27, 2197-2200, 10.1029/1999gl010923, 2000.

Garrett, T. J., and Zhao, C.: Increased Arctic cloud longwave emissivity associated with pollution from mid-latitudes, Nature, 440, 787-789, 10.1038/nature04636, 2006.

Guirado-Fuentes, C., Cuevas, E., Cachorro, V., Toledano, C., Alonso-Pérez, S., Bustos, J., Basart, S., Romero, P., Camino, C., Mimouni, M., Zeudmi, L., Goloub, P., Baldasano, J., and Frutos Baraja, A.: Aerosol characterization at the Saharan AERONET site Tamanrasset, Atmospheric Chemistry and Physics, 14, 11753-11773, 10.5194/acp-14-11753-2014, 2014.

Li, L., Li, Z., Chang, W., Ou, Y., Goloub, P., Li, C., Li, K., Hu, Q., Wang, J., and Wendisch, M.: Aerosol solar radiative forcing near the Taklimakan Desert based on radiative transfer and regional meteorological simulations during the Dust Aerosol Observation-Kashi campaign, Atmos. Chem. Phys., 20, 10845-10864, 10.5194/acp-20-10845-2020, 2020.

Loeb, N., Wielicki, B., Hu, Y., Iii, J., and Stowe, L.: Clouds and the Earth's Radiant Energy System (CERES) Validation Plan CERES Inversion to Instantaneous TOA Fluxes (Subsystem 4.5), 2021.

McComiskey, A., Ricchiazzi, P., Ogren, J., and Dutton, E.: SGPGET: AN SBDART Module for Aerosol Radiative Transfer, 2021.

Mishchenko, M. I., and Travis, L. D.: Capabilities and limitations of a current FORTRAN implementation of the T-matrix method for randomly oriented, rotationally symmetric scatterers,

Journal of Quantitative Spectroscopy and Radiative Transfer, 60, 309-324, https://doi.org/10.1016/S0022-4073(98)00008-9, 1998.

Ramanathan, V., Cess, R., Harrison, E., Minnis, P., Barkstrom, B., Ahmad, E., and Hartmann, D.: Cloud-Radiative Forcing and Climate: Results from The Earth Radiation Budget Experiment, Science (New York, N.Y.), 243, 57-63, 10.1126/science.243.4887.57, 1989.

Ricchiazzi, P., Yang, S., Gautier, C., and Sowle, D.: SBDART: A Research and Teaching Software Tool for Plane-Parallel Radiative Transfer in the Earth's Atmosphere, Bulletin of the American Meteorological Society, 79, 2101, 10.1175/1520-0477(1998)079<2101:Sarats>2.0.Co;2, 1998.

Sayer, A. M., Munchak, L. A., Hsu, N. C., Levy, R. C., Bettenhausen, C., and Jeong, M.-J.: MODIS Collection 6 aerosol products: Comparison between Aqua's e-Deep Blue, Dark Target, and "merged" data sets, and usage recommendations, Journal of Geophysical Research: Atmospheres, 119, 13,965-913,989, 10.1002/2014jd022453, 2014.

Schaaf, C., Martonchik, J., Pinty, B., Govaerts, Y., Gao, F., Lattanzio, A., Liu, J., Strahler, A., and Taberner, M.: Retrieval of Surface Albedo from Satellite Sensors, in: Advances in Land Remote Sensing: System, Modeling, Inversion and Application, edited by: Liang, S., Springer Netherlands, Dordrecht, 219-243, 2008.

Schaaf, C. B., Gao, F., Strahler, A. H., Lucht, W., Li, X., Tsang, T., Strugnell, N. C., Zhang, X., Jin, Y., Muller, J.-P., Lewis, P., Barnsley, M., Hobson, P., Disney, M., Roberts, G., Dunderdale, M., Doll, C., d'Entremont, R. P., Hu, B., Liang, S., Privette, J. L., and Roy, D.: First operational BRDF, albedo nadir reflectance products from MODIS, Remote Sensing of Environment, 83, 135-148, https://doi.org/10.1016/S0034-4257(02)00091-3, 2002.

Schaaf, C. B., Liu, J., Gao, F., and Strahler, A. H.: Aqua and Terra MODIS Albedo and Reflectance Anisotropy Products, in: Land Remote Sensing and Global Environmental Change: NASA's Earth Observing System and the Science of ASTER and MODIS, edited by: Ramachandran, B., Justice, C. O., and Abrams, M. J., Springer New York, New York, NY, 549-561, 2011.

Su, W., Corbett, J., Eitzen, Z., Liang, and L.: Next-generation angular distribution models for top-of-atmosphere radiative flux calculation from the CERES instruments: validation, Atmospheric Measurement Techniques Discussions, 2015.

Tian, L., Zhang, P., and Chen, L.: Estimation of the Dust Aerosol Shortwave Direct Forcing Over Land Based on an Equi-albedo Method From Satellite Measurements, Journal of Geophysical Research: Atmospheres, 124, 8793-8807, 10.1029/2019JD030974, 2019.

Zhang, J., Christopher, S. A., Remer, L. A., and Kaufman, Y. J.: Shortwave aerosol radiative forcing over cloud-free oceans from Terra: 2. Seasonal and global distributions, Journal of Geophysical Research (Atmospheres), 110, D10S24, 2005.

---

## Author Response (AR2)

**Responses to the comments**

Dear Editor,

Thank you for handling and carefully reviewing our manuscript. The comments and suggestions would helpe us to improve the paper.

Following these comments and suggestions, we take a lot of efforts to revised the manuscript, re-draw the most of the imaging in the article, and modified the inaccurate information to avoid the ambiguity.

We response every comments sentence by sentence. Please find the comments in blue italics and our reply in black.

**Comments**

1. You do not need the same 3 colour bars per line, just use a single one since they repeat themselves. In addition if you were to put the 1 colour bar on the side of the Figures you would be able to expand there size. I recommend that you make the numbers on these colour bars more legible (larger font). Please apply these recommendations to ALL Figures: 3, 4, 5, 6 and 8 & 9. When colour bars are all the same on the Figure, I expect to see a UNIQUE colour bar.

**Reply:** Thank you for the valuable suggestion, we re-drawed the Figures 3, 4, 5, 6 and 8 & 9, using single colour bar and make the numbers on these colour bars more legible. Please see Fig.3 in Page 10, Fig.4 in Page 11, Fig.5 in Page 13, Fig.6 in Page 14, Fig.8 in Page 17 and Fig.9 in Page 18 of the revised manuscript.

[Figure]

**Figure 3: SW LSA and SZA over the Sahara Desert and the Taklimakan Desert derived from AQUA/MODIS.**

[Figure]

**Figure 4: True color images and cloud detections from AQUA/MODIS observations.**

[Figure]

**Figure 5: TOA SW radiative flux derived from AQUA/CERES over the Sahara Desert on March 2019 and over the Taklimakan Desert on April 2019.**

[Figure]

**Figure 6: AOD and DRE$_{dust}$ of dust storms over the Sahara Desert in March 2019 and over the Taklimakan Desert in April 2019.**

[Figure]

**Figure 8: Integrated water vapor (g/cm²) from European Centre for Medium-range Weather Forecasts (ECMWF) reanalysis dataset over the Sahara Desert in March 2019 and over the Taklimakan Desert in April 2019.**

[Figure]

**Figure 9:** **SBDART simulated clear-sky TOA radiative flux by using integrated water vapor (g/cm$^2$) from ECMWF reanalysis dataset over the Sahara Desert in March 2019 and over the Taklimakan Desert in April 2019.**

2.  The aerosols measure in Tamanrassett DO NOT represent the dust aerosol from Sahara due to the variations in mineralogical composition of soil over the region. Please make that clear in your text. The same would apply for Kashi with respect to the Taklamakan Desert.

**Reply:** Thank you for your reminding. It is ture that the dust aerosol properties derived from single site measurements could not represent the whole source region due to the variations in mineralogical composition of soil over the region. The large spatial variability of aerosols and the lack of an adequate database on their properties make DRE$_{dust}$ and DRFE$_{dust}$ much very difficult to be estimated (Satheesh and Srinivasan, 2006). In this paper, we using a satellite-based method to estimate the DRFE$_{dust}$ over land without any assumptions of the dust aerosol properties to overcame the problem successfully and got a good result. In order to evaluate DRE$_{dust}$ and DRFE$_{dust}$ accurately, more ground observations are needed for represents the dust aerosol properties variation in the whole source region in detail. However, there can not have so many ground observation sites in practice. Therefore, some previous

studies using dust aerosol properties derived from single site to represents dust aerosol properties of the research area (Li et al., 2020;Guirado-Fuentes et al., 2014;Garcń et al., 2014;Garcń et al., 2012).

In the paper, we chose the satellite data around Tamanrasset and Kashi to estimate the $DRFE_{dust}$, and we consider the slope of the linear regression line through these data points (Fig.7) could presents the mean $DRFE_{dust}$ around Tamanrasset and Kashi. We have modified the inaccurate information to avoid the ambiguity. Please see Lines 26-30 in Page1, Lines 92-93 in Page 4, Lines 216-220 in Page 9, Lines 407-409 and Lines 413-417 in Page 21, Lines 424-426 in Page 22, Lines 441-443 and Lines 449-453 in Page 23, Lines 461-464 in Page 24, Lines 467-469 in Page 25 and Lines 527-530 in Page 28 of the revised manuscript.

3. The following is incorrect:
14. Line 257: "Thus, dust aerosols have a negative radiative effect in the SW spectrum." ...Here again --how do I get this from Fig 6, where all numbers are positive? If you want to discuss radiative effect, why not show radiative effect in the figure? ...I see you have rad effect in Fig 7. Why not delay this discussion until then?

Reply: Dust aerosols have higher SW albedo than land surface albedo in clear-sky conditions, and dust aerosols reflect more SW radiation to TOA. DREdust was defined as the radiative fluxes difference between clear (Fclr) and dust loading (Fdust) conditions (Garrett and Zhao, 2006; Christopher et al., 2000;Ramanathan et al., 1989).
DRE dust = Fclr - Fdust

Therefore dust aerosols have a negative radiative effect in the SW spectrum. It also can be founded in Fig. 6 in the revised manuscript, we delayed this discussion behind Fig. 6 in the revised manuscript. Please see Lines 312-313 in Page 15 of the revised manuscript.

Dust aerosols can either have a positive or a negative radiative effect at the TOA as discussed by many authors (Sokolik and Toon (1999); Claquin et al., 1999; Liu and Seinfeld (1999) Balkanski et al.(2007) Miller et al. (2004 and 2011)…. If you were talking about the surface radiative effect in clear-sky then you would be right, it is always negative.

Also, please check your definition of DRE dust, is it 'Fclr – Fdust' or 'Fdust- Fclr' that you are discussing?

**Reply:** $DRE_{dust}$ is defined as the upward radiative flux difference between clear ($F_{clr}$) and dust loading ($F_{dust}$) conditions ($DRE_{dust}=F_{clr} - F_{dust}$). And here we discuses the shortwave (SW) $DRE_{dust}$ at the top of the atmosphere (TOA) in this paper. Dust aerosols have higher SW albedo than land surface in clear-sky conditions, and dust aerosols reflect more SW radiation to TOA, therefore dust aerosols causing negative SW $DRE_{dust}$ at the TOA. Previous studies also show dust causes SW negative radiative effect at the TOA (Xia and Zong, 2009;Tian et al., 2019;Lin et al., 2009;Li et al., 2020;Li et al., 2004;Jose et al., 2016;Garcń et al., 2014;Garcń et al., 2012;Christopher and Zhang, 2002;Bi et al., 2014).

4. Please correct this sentence as I could not figure out what you meant in lines 379-380: " However, the contents of the SW radiative flux change little with the increase of the height 379 of dust layer (within 1.5Wm-2, 0.47%). "

**Reply:** It has been rewritten as "However, the SW radiative flux change little (within 1.5Wm$^{-2}$, 0.47%) with the increase of the height of dust layer.". Please see Lines 379-380 in Page 19 of the revised manuscript.

5. There is no such thing as a dust aerosol typical of the Sahara since it is mineralogy that mostly determines optical properties. The same is true for the Taklimakan Desert. If you convey this message anywhere in the paper, it should be edited.

**Reply:** Thank you for the valuable suggestion. Following the suggestion, we have modified the inaccurate information to avoid the ambiguity. Please see Lines 26-30 in Page1, Lines 92-93 in Page 4, Lines 216-220 in Page 9, Lines 407-409 and Lines 413-417 in Page 21, Lines 424-426 in Page 22, Lines 441-443 and Lines 449-453 in Page 23, Lines 461-464 in Page 24, Lines 467-469 in Page 25 and Lines 527-530 in Page 28 of the revised manuscript.

6. This sentence needs to be modified to better reflect what you mean to say:
 "Therefore, the uncertainties can be evaluated more reasonably."

**Reply:** It has been rewritten as "Therefore, the uncertainties can be estimated objectively.". Please see Line 519 in Page 28 of the revised manuscript.

**References**

Bi, J., Shi, J., Xie, Y., Liu, Y., Takamura, T., and Khatri, P.: Dust Aerosol Characteristics and Shortwave Radiative Impact at a Gobi Desert of Northwest China during the Spring of 2012, Journal of the Meteorological Society of Japan. Ser. II, 92A, 33-56, 10.2151/jmsj.2014-A03, 2014.

Christopher, S. A., and Zhang, J.: Shortwave Aerosol Radiative Forcing from MODIS and CERES observations over the oceans, Geophysical Research Letters, 29, 6-1-6-4, 10.1029/2002gl014803, 2002.

García, O. E., Díaz, J. P., Expósito, F. J., Díaz, A. M., Dubovik, O., Derimian, Y., Dubuisson, P., and Roger, J. C.: Shortwave radiative forcing and efficiency of key aerosol types using AERONET data, Atmos. Chem. Phys., 12, 5129-5145, 10.5194/acp-12-5129-2012, 2012.

García, R. D., García, O. E., Cuevas, E., Cachorro, V. E., Romero-Campos, P. M., Ramos, R., and de Frutos, A. M.: Solar radiation measurements compared to simulations at the BSRN Izaña station. Mineral dust radiative forcing and efficiency study, Journal of Geophysical Research: Atmospheres, 119, 179-194, https://doi.org/10.1002/2013JD020301, 2014.

Guirado-Fuentes, C., Cuevas, E., Cachorro, V., Toledano, C., Alonso-Pérez, S., Bustos, J., Basart, S., Romero, P., Camino, C., Mimouni, M., Zeudmi, L., Goloub, P., Baldasano, J., and Frutos Baraja, A.: Aerosol characterization at the Saharan AERONET site Tamanrasset, Atmospheric Chemistry and Physics, 14, 11753-11773, 10.5194/acp-14-11753-2014, 2014.

Jose, S., Gharai, B., Rao, P. V. N., and Dutt, C. B. S.: Satellite-based shortwave aerosol radiative forcing of dust storm over the Arabian Sea, Atmospheric Science Letters, 17, 43-50, https://doi.org/10.1002/asl.597, 2016.

Li, F., Vogelmann, A. M., and Ramanathan, V.: Saharan Dust Aerosol Radiative Forcing Measured from Space, Journal of Climate, 17, 2558-2571, 10.1175/1520-0442(2004)017<2558:SDARFM>2.0.CO;2, 2004.

Li, L., Li, Z., Chang, W., Ou, Y., Goloub, P., Li, C., Li, K., Hu, Q., Wang, J., and Wendisch, M.: Aerosol solar radiative forcing near the Taklimakan Desert based on radiative transfer and regional meteorological simulations during the Dust Aerosol Observation-Kashi campaign, Atmos. Chem. Phys.,

20, 10845-10864, 10.5194/acp-20-10845-2020, 2020.

Lin, C., Guang-Yu, S., Ling-Zhi, Z., and Sai-Chun, T.: Assessment of Dust Aerosol Optical Depth and Shortwave Radiative Forcing over the Northwest Pacific Ocean in Spring Based on Satellite Observations, Atmospheric and Oceanic Science Letters, 2009.

Satheesh, S. K., and Srinivasan, J.: A Method to Estimate Aerosol Radiative Forcing from Spectral Optical Depths, Journal of Atmospheric Sciences, 63, 1082, 10.1175/jas3663.1, 2006.

Tian, L., Zhang, P., and Chen, L.: Estimation of the Dust Aerosol Shortwave Direct Forcing Over Land Based on an Equi-albedo Method From Satellite Measurements, Journal of Geophysical Research: Atmospheres, 124, 8793-8807, 10.1029/2019JD030974, 2019.

Xia, X., and Zong, X.: Shortwave versus longwave direct radiative forcing by Taklimakan dust aerosols, Geophysical Research Letters, 36, 10.1029/2009gl037237, 2009.

---

## Author Response (AR3)

**Responses to the comments**

Dear Editor,

Thank you very much for helping us to improve the manuscript. Following the suggestion, we have the corrected the sentence. Please see Lines 382-383 in Page 19 of the revised manuscript.

Many thanks and best regards,

Lin Tian